# The IL6/JAK/STAT3 signaling axis is a therapeutic vulnerability in SMARCB1-deficient bladder cancer

Chandra Sekhar Amara [1], Karthik Reddy Kami Reddy[1,2,18], Yang Yuntao[3,18], Yuen San Chan[1,2,4], Danthasinghe Waduge Badrajee Piyarathna[1], Lacey Elizabeth Dobrolecki [5,6], David J. H. Shih [3], Zhongcheng Shi[5], Jun Xu[1,5], Shixia Huang[1,2,5,7], Matthew J. Ellis[6], Andrea B. Apolo [8], Leomar Y. Ballester[9], Jianjun Gao [10], Donna E. Hansel[9], Yair Lotan [11], H. Courtney Hodges [1,2,4,12], Seth P. Lerner [13], Chad J. Creighton [2,14], Arun Sreekumar[1,2], W. Jim Zheng [3], Pavlos Msaouel [15,16,17] ✉, Shyam M. Kavuri [6] ✉ & Nagireddy Putluri [1,2,5] ✉

SMARCB1 loss has long been observed in many solid tumors. However, there is a need to elucidate targetable pathways driving growth and metastasis in SMARCB1-deficient tumors. Here, we demonstrate that SMARCB1 deficiency, defined as genomic SMARCB1 copy number loss associated with reduced mRNA, drives disease progression in patients with bladder cancer by engaging STAT3. SMARCB1 loss increases the chromatin accessibility of the STAT3 locus in vitro. Orthotopically implanted SMARCB1 knockout (KO) cell lines exhibit increased tumor growth and metastasis. SMARCB1-deficient tumors show an increased IL6/JAK/STAT3 signaling axis in in vivo models and patients. Furthermore, a pSTAT3 selective inhibitor, TTI-101, reduces tumor growth in SMARCB1 KO orthotopic cell line-derived xenografts and a SMARCB1-deficient patient derived xenograft model. We have identified a gene signature generated from SMARCB1 KO tumors that predicts SMARCB1 deficiency in patients. Overall, these findings support the clinical evaluation of STAT3 inhibitors for the treatment of SMARCB1-deficient bladder cancer.

Bladder cancer (BLCA) is a common malignancy that causes more than 150,000 deaths per year worldwide[1]. It is the fourth most common cancer in men and ninth overall among other cancer types[2]. So far, there are only a few FDA-approved molecularly targeted agents for the treatment of metastatic BLCA[3–5].

BLCA represents a broad spectrum of diseases, from low-risk, low-grade non-muscle invasive to high-grade muscle invasive cancers to locally advanced unresectable or, metastatic disease[6]. One of the challenging aspects of clinically managing BLCA is its heterogeneity with respect to invasion and formation of new or recurring tumors in the bladder. Deciphering the molecular mechanisms involved in the development of BLCA metastasis is essential for developing effective tumor-specific therapies.

The genomic landscape of patients with BLCA revealed that >60% of cases have inactivation in the components of the SWItch/Sucrose Non-Fermentable (SWI/SNF) nucleosome remodeling family of complexes[7]. SWI/SNF complexes are comprised of multiple subunits, including SMARCB1, SMARCA4, ARID1A, ARID1B, PBRM1, BRD7, SMARCC1, and SMARCC2[8,9]. The SMARCB1 (SWI/SNF-Related Matrix-Associated Actin-Dependent Regulator of Chromatin Subfamily B

Member 1) subunit, also known as integrase interactor 1 (INI-1), or BRG1 associated factor 47 (BAF47), is critical for the chromatin remodeling function of SWI/SNF complexes[10–15].

Recent data suggest that low SMARCB1 expression facilitates BLCA growth via the activation of STAT3[16]. This is corroborated by the notable STAT3 pathway upregulation found in renal medullary carcinoma, a highly aggressive renal cell carcinoma subtype characterized by the loss of SMARCB1[17]. However, the molecular mechanisms behind these observations remain to be elucidated, and the utility of STAT3 pathway targeting has not been investigated in SMARCB1-deficient tumors. In the present study, we hypothesized that targeting STAT3 would be most efficacious in BLCA tumors harboring SMARCB1 deficiency defined as SMARCB1 copy number loss associated with decreased SMARCB1 mRNA expression. To test this hypothesis, we sought to elucidate the impact of SMARCB1 loss and subsequent STAT3 pathway activation on bladder orthotopic tumor growth, metastasis, and BLCA disease-specific survival. Additionally, we examined the molecular mechanisms of metastasis and examined therapeutic strategies to overcome tumor growth and metastasis in SMARCB1-deficient BLCA driven by STAT3 pathway upregulation. Further, we established a transcriptional signature, which can be used to identify subgroups of patients with SMARCB1-deficient BLCA. These findings could provide preclinical rationale for effectively treating this subset of patients with SMARCB1-deficient BLCA.

## Results

### Low SMARCB1 expression in BLCA tumors is associated with worse patient outcomes and IL6/JAK/STAT3 pathway upregulation

To obtain better insights on individual SWI/SNF components in BLCA, we assessed the association of survival with the mRNA expression of each of these subunits (low vs high) using maximally selected rank statistics[18] from The Cancer Genome Atlas Urothelial Bladder carcinoma (TCGA-BLCA) cohort (refer to methods section) [7]. We found that low expression of SMARCB1 ($p = 0.031$) (Fig. 1A; Supplementary Data 1A), ARID1A ($p = 0.047$)[19], and SMARCD1 ($p = 0.047$) (Supplementary Fig. 1) were significantly associated with worse survival outcomes. We subsequently focused our investigation for this study on SMARCB1 because it showed the strongest association with survival and is an established tumor suppressor and key driver of other genitourinary malignancies such as Renal Medullary Carcinoma (RMC)[17]. We performed gene set enrichment analysis (GSEA) to identify the pathways significantly enriched in BLCA tumors harboring low SMARCB1 expression. Hallmark signaling pathways associated with inflammatory responses and IL6/JAK/STAT3 signaling pathways were among the top enriched gene sets associated with low SMARCB1 expression (Fig. 1B, C; Supplementary Data 2).

To verify these findings, we performed transcriptomic analysis in additional previously published BLCA datasets[20–22], and again consistently found that SMARCB1 low tumors were significantly enriched for the IL6/JAK/STAT3 pathway (Supplementary Fig. 2A–C; Supplementary Data 3). The lowest SMARCB1 levels in the TCGA-BLCA cohort were noted in the subset of tumors that harbored either deep or shallow SMARCB1 deletions (~34% of BLCA tumors) (Fig. 1D). Furthermore, GSEA analysis of the TCGA-BLCA cohort demonstrated significant enrichment of the IL6/JAK/STAT3 signaling pathway in tumors harboring either deep or shallow SMARCB1 deletions compared with SMARCB1 diploid tumors (Supplementary Fig. 2D; Supplementary Data 4). In contrast to SMARCB1 shallow/deep deletions, the frequency of SMARCB1 mutations in BLCA was very low ($n = 5$ out of 402; Supplementary Fig. 2E). Broader interrogation of the TCGA pan-cancer data (excluding BLCA) revealed that ~24% of tumors harbor deep or shallow deletions associated with low SMARCB1 mRNA expression (Supplementary Fig. 2F; Supplementary Data 5). Further, we categorized BLCA tumors into six different groups based on the presence of SMARCB1

genomic alterations (shallow/deep deletions, diploid, gain, and amplification) along with SMARCB1 mRNA expression levels. We observed that only group I (SMARCB1 shallow/deep deletions and low mRNA), which we referred hereafter as the "SMARCB1-deficient" BLCA cohort, showed significant increase of pSTAT3 as compared to either diploid or gain of SMARCB1 BLCA cohorts (Fig. 1E; Supplementary Data 1B).

### Loss of SMARCB1 increases tumor growth and metastasis

To investigate the role of SMARCB1 loss in mediating tumor growth and metastasis, we used CRISPR/Cas9 to knockout (KO) SMARCB1 (from clone C16) in the human T24 BLCA cell line (Supplementary Fig. 3). As shown in Fig. 2A, our immunoblot analysis confirmed depletion of SMARCB1 protein in KO cell lines compared to respective controls expressing non-targeting sgRNA (empty vector). To validate the specificity of the KO phenotype, we generated a SMARCB1 rescue cell line by re-expressing the full-length SMARCB1 in the KO background cells as confirmed by immunoblot analysis (Fig. 2A). All the cells were luciferase labeled to facilitate Bioluminescence imaging (BLI) in vivo. To investigate whether loss of SMARCB1 promotes BLCA disease progression, we used an orthotopic mouse NOD/SCID/IL2rγ-null (NSG) model wherein luciferase labeled T24 human bladder cancer cells (control, KO, or rescue) were directly injected into the bladder wall (orthotopic site) using two independent KO (SMARCB1 loss) clones (C16 and C45) and respective SMARCB1 rescue lines. We monitored orthotopic tumor growth using BLI signal. Within 2–3 weeks post-injection, we observed significantly increased BLI signal in the bladder region of mice implanted with SMARCB1 KO (clone C16) cells compared with T24 control cells (Fig. 2B; Supplementary Fig. 4A). Mice implanted with rescue cells had similar levels of luciferase signal to control cells, ruling out the possibility that the observed phenotype in KO cells was due to off-target effects. At the endpoint (-week 3) the weight of KO xenografts tumors was >10-fold compared to controls (Fig. 2C). The increased tumor growth in SMARCB1 KO was evidently abrogated in rescue cells, wherein SMARCB1 expression was similar to control (Fig. 2C). This observation was further confirmed in the independent SMARCB1 KO clone (C45) derived from the T24 BLCA cell lines (Supplementary Fig. 4B). Tumors from these cells also showed >10-fold increase in the KO compared to the control or rescue (Supplementary Fig. 4C).

SMARCB1 functions in the stabilization of the SWI/SNF complex, and therefore loss of this gene leads to instability of the SWI/SNF complex regulation of the promoter/enhancer regions of the target genes[23]. To determine whether SMARCB1 KO affected the expression of SWI/SNF complex subunits, we measured the endogenous protein expression of core complex genes SMARCA4, SMARCC1, and SMARCC2 in both T24 SMARCB1 KO and rescue cells from clone C16. Our protein analysis did not reveal any changes to the other core subunits upon KO of SMARCB1, suggesting that the observed phenotype is specific to the loss of SMARCB1 (Supplementary Fig. 4D). We successfully generated SMARCB1 knockdown (KD) using two independent shRNAs to generate T24 SMARCB1 KD cell lines (KD1 and KD2) and confirmed the decrease in SMARCB1 mRNA and protein expression levels compared with SMARCB1 KO T24 cell lines (Supplementary Fig. 4E, F). We further investigated the effect of tumor growth using these cell lines. Our orthotopic xenograft studies revealed that both KDs and KO (clone 16) showed increased tumor growth, but KO had significant effect of tumor growth compared to KDs (Supplementary Fig. 4G). To validate our findings, we used the luciferase labeled T24 cells from the C16 clone (control, KO, rescue) to perform spheroid assays and observed increased spheroid growth in SMARCB1 KO compared to control and rescue (Supplementary Fig. 5).

To further assess the role of SMARCB1 in metastatic progression, we harvested and analyzed BLI signal for lungs, liver, kidneys, and stomach & intestines from each mouse by ex-vivo. All ex-vivo lungs, liver, and stomach & intestines from mice bearing SMARCB1 KO cells

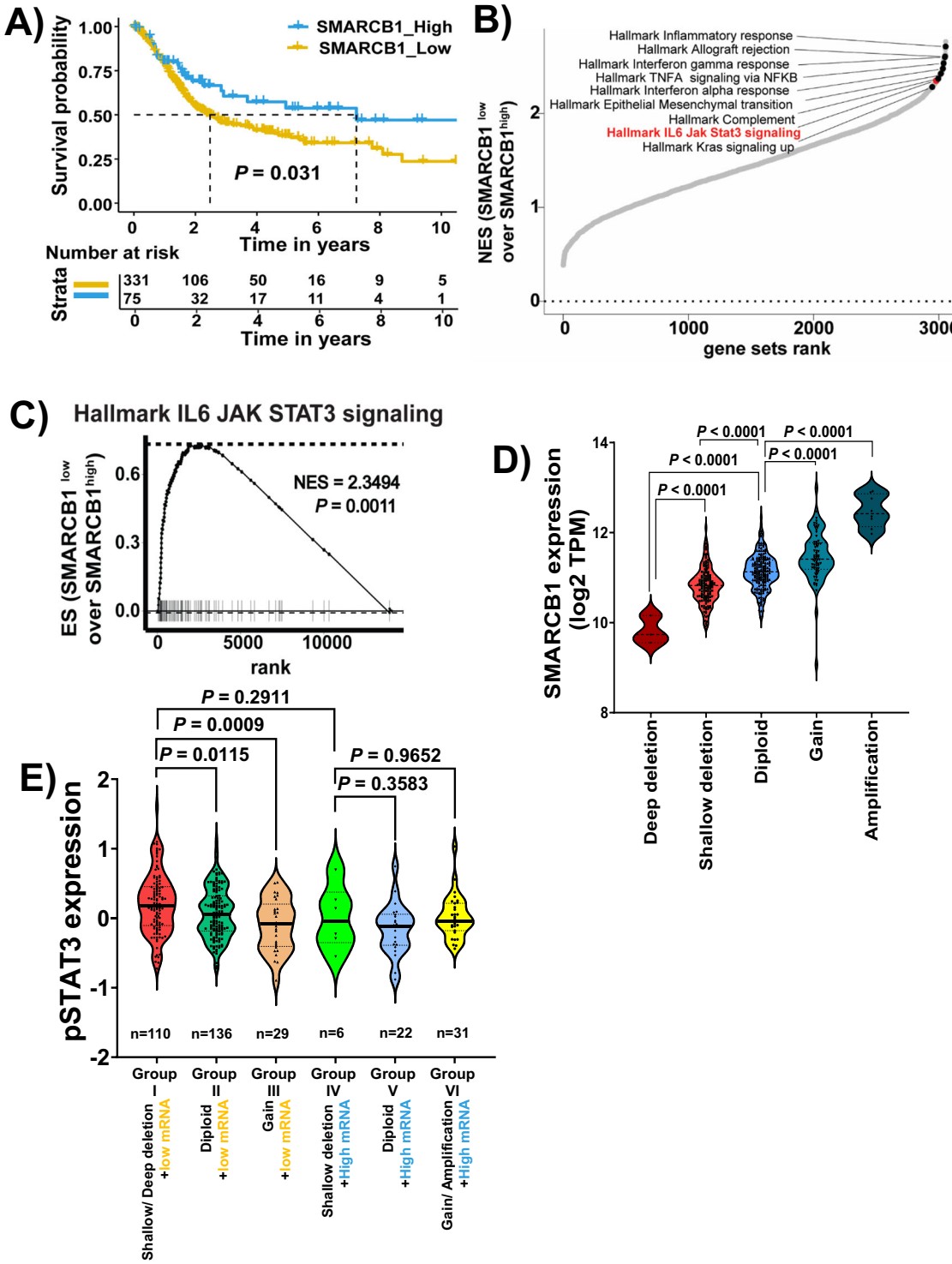

exhibited the BLI signal, but only a few ex-vivo kidneys exhibited the BLI signal (Supplementary Fig. 6A). BLI signal and total flux in organs harvested from mice bearing SMARCB1 KO xenografts were increased compared to mice bearing control and rescue xenografts (Fig. 2D, E). Hematoxylin and eosin (H&E) analysis of the lungs, liver, kidneys, and intestine sections from SMARCB1 KO xenografts confirmed the metastatic potential of SMARCB1 KO orthotopic tumors (Fig. 2F). Immunohistochemistry (IHC) analyses for SMARCB1 on the sections derived from metastatic lesions also validated that these metastatic lesions are derived from SMARCB1 KO orthotopic tumors, suggesting that loss of SMARCB1 increased BLCA metastasis (Supplementary Fig. 6B), whereas the organs derived from mice bearing control and

SMARCB1 rescue xenografts did not show any presence of metastatic lesions (Supplementary Fig. 7A). Overall, these in vivo experimental findings suggest that loss of SMARCB1 increased BLCA tumor growth and metastasis, consistent with the clinical association of low SMARCB1 BLCA tumors with worse outcomes.

To validate these findings in a clinical setting, we evaluated primary and matched metastatic tissues from seven patients with BLCA treated in a Phase II clinical trial (NCT02788201) from the National Cancer Institute (NCI). We found decreased mRNA expression of SMARCB1 in the metastatic lesions compared to matched primary tumors indicating that low SMARCB1 levels are associated with distant metastases (Supplementary Fig. 7B).

**Fig. 1 | SMARCB1 deficiency is associated with worse outcomes and enrichment for STAT3 signaling. A** Low expression of SMARCB1 was associated with worse survival in TCGA-BLCA. Kaplan–Meier plot with bladder cancer ($n = 406$) cohort defined by low ($n = 331$) or high ($n = 75$) SMARCB1 mRNA expression based on maximally selected rank statistics [log-rank test ($P = 0.031$; two-sided); only patients with survival information were represented]. **B** Ranking of hallmark gene sets that are enriched in low SMARCB1 ($n = 331$) compared to high SMARCB1 ($n = 75$) based on maximally selected rank statistics in TCGA-BLCA patients ($n = 406$). **C** Gene set enrichment analysis (GSEA) plot for the HALLMARK IL6/JAK/STAT3 signaling pathway comparing SMARCB1 low ($n = 331$) with SMARCB1 high ($n = 75$) in TCGA-BLCA patients ($n = 406$) (NES = 2.3494; $P = 0.0011$). GSEA analysis was performed using 1000 permutations to determine a significance $p$-value. **D** Association between SMARCB1 mRNA expression and SMARCB1 copy number alterations including deep deletion ($n = 3$); shallow deletion ($n = 138$), diploid ($n = 184$), gain ($n = 69$); amplification ($n = 8$) (4 patients' copy number alteration data was unavailable) in the TCGA-BLCA cohort. Violin plots represent the expression levels of SMARCB1 mRNA expression with respect to copy number alterations of SMARCB1 in the TCGA-BLCA patient cohort. **E** Violin plot showing the correlation between pSTAT3 (Y705) levels (RPPA-TCGA) and SMARCB1 in the TCGA-BLCA cohort. SMARCB1 mRNA levels were defined based on KM plot in Fig.1A. The patients were divided into six groups by considering SMARCB1 copy number alterations and SMARCB1 mRNA levels (Group I: SMARCB1 shallow/deep deletion with low SMARCB1 mRNA; Group II: SMARCB1 diploid with low SMARCB1 mRNA; Group III: SMARCB1 gain with low SMARCB1 mRNA; Group IV: SMARCB1 shallow deletion with high SMARCB1 mRNA; Group V: SMARCB1 diploid with high SMARCB1 mRNA; Group VI: SMARCB1 gain/ amplification with high SMARCB1 mRNA) [Note: pSTAT3 (Y705) data were obtained from RPPA-TCGA and available for matched patients with copy number alterations and survival ($n = 334$ out of 406 used for analysis, and refer to "Methods" section)]. For panels **D** and **E**, $P$-values were determined by unpaired two-tailed Student's $t$-test. Source data are provided as a Source Data file.

## SMARCB1 loss leads to activation of IL6/JAK/STAT3 signaling

To identify SMARCB1-specific targets enriched upon SMARCB1 loss (KO) and gain (rescue of SMARCB1), we performed transcriptomic profiling (RNA-seq) from T24 Control, SMARCB1 KO (clone 16) and SMARCB1 rescue orthotopic xenografts using both Salmon and XenofilteR methods. We performed GSEA comparing SMARCB1 KO over control and SMARCB1 rescue over KO to identify pathways enriched by SMARCB1 loss. IL6/JAK/STAT3 signaling was one of the top most positively enriched pathways in SMARCB1 KO (Fig. 3A; Supplementary Fig. 8A–F; Supplementary Data 6) and this pathway was negatively enriched upon rescue of SMARCB1 (Fig. 3B; Supplementary Fig. 8A–F; Supplementary Data 6).

Since our transcriptomic comparisons consistently showed enrichment of the IL6/JAK/STAT3 pathway in SMARCB1-deficient BLCA tumors, we further verified the transcript levels of IL6, JAK1, and STAT3 by qRT-PCR from orthotopic tumors. Our analysis revealed significantly increased mRNA expression of STAT3, IL6 and JAK1 in SMARCB1 KO T24 xenografts (clone 16) which were decreased upon rescue of SMARCB1 (Fig.3C; Supplementary Fig. 9A, B). In the same setting, immunoblot analysis revealed significantly increased STAT3 and pSTAT3 (Y705) protein levels in SMARCB1 KO compared with SMARCB1 control and rescue tumors (Fig. 3D). Furthermore, immunoblot analysis in T24 spheroids revealed a significant increase in the total JAK1 protein expression (Supplementary Fig. 9C). However, we were unable to detect phosphorylated form of JAK1 (pJAK1) by immunoblot analysis. Thus, we performed Reverse Phase Protein Array (RPPA) analysis for pJAK1 in SMARCB1 KO compared with control and SMARCB1 rescue spheroids, and the results revealed a significant increase in pJAK1 levels in SMARCB1 KO spheroids (Supplementary Fig. 9D). We subsequently measured the IL6 levels in supernatants from control, KO and rescue derived cell lines and found increased IL6 levels in KO cell supernatants compared to SMARCB1 control and rescue cell supernatants (Supplementary Fig. 9E). We notably observed increased levels of IL6 in mouse plasma isolated from mice bearing SMARCB1 KD and KO xenografts (Supplementary Fig. 9F). Further, we observed increased levels of pSTAT3 (Y705) and STAT3 in both KD and KO xenograft tumors (Supplementary Fig. 9G).

To investigate the chromatin changes induced by SMARCB1 loss in BLCA cells, we performed an assay for transposase-accessible chromatin with sequencing (ATAC-seq) analysis on SMARCB1 KO, control, and rescue BLCA cell lines (Supplementary Fig. 10A–F). We noted 53952 overlapping peaks that were decreased in SMARCB1 KO over control and increased in SMARCB1 rescue over SMARCB1 KO. Moreover, 30172 overlapping peaks were increased in SMARCB1 KO over control and decreased in SMARCB1 rescue over SMARCB1 KO (Supplementary Fig. 10B). Mapping of these peaks to hallmark pathways identified IL6/JAK/STAT3 signaling pathway as one of the pathways demonstrating increased accessibility upon SMARCB1 KO

(Supplementary Fig. 10C; Supplementary Data 7). Further transcriptomic motif enrichment analysis shows an increase of STAT3 accessibility upon SMARCB1 KO (Supplementary Fig. 10D; Supplementary Data 8). SMARCB1 KO cells demonstrated increased accessibility of the STAT3 and IL6 loci as compared to control or SMARCB1 rescue (Supplementary Fig. 10E, F). These results suggest that SMARCB1 may repress STAT3 pathway expression by affecting the chromatin accessibility. To verify this, we performed chromatin immunoprecipitation (ChIP) followed by quantitative real-time PCR (qPCR) on the STAT3 promoter locus relative to a negative control region of no enrichment (Fig. 3E; Supplementary Fig. 10G). Our results revealed enrichment of histones H3K27ac and H3K4me3 at the STAT3 promoter regions in SMARCB1 KO compared to control and rescue T24 cell line-derived orthotopic xenografts (Fig. 3E). We further verified the binding of SMARCB1 to the STAT3 promoter by ChIP-qPCR and observed decreased SMARCB1 enrichment in KO which was rescued upon rescue of SMARCB1 (Fig. 3F). Similarly, ChIP-qPCR in SMARCB1 knockdown (KD2) cell line-derived orthotopic xenografts showed significantly increased accessibility at the STAT3 promoter compared to control derived xenografts (Supplementary Fig. 10G). These mechanistic findings suggest that SMARCB1 deficiency directly deregulates STAT3 expression. To further investigate the upstream target that drives the increased pSTAT3 (Y705) levels observed in SMARCB1 KO BLCA cells, we treated spheroids with Itacitinib, a specific JAK1 inhibitor[24,25], that abrogated pSTAT3 (Y705) levels in SMARCB1 KO derived spheroids (Fig. 3G).

## STAT3 is critical for SMARCB1-deficient BLCA tumor growth and metastasis

To understand the role of STAT3 in SMARCB1-deficient BLCA progression, we generated KD of STAT3 using shRNA in T24 BLCA cells using both the control and SMARCB1 KO (using clone C16) (Supplementary Fig. 11A; Fig. 4A). These STAT3 KD in SMARCB1 KO and corresponding control cells were engrafted into orthotopic xenografts using NSG mice. We found that STAT3 KD reduced tumor growth (Fig. 4B, C) and lung metastasis (Fig. 4D) compared with STAT3 control in SMARCB1 KO BLCA tumors. Furthermore, in vitro spheroid assays demonstrated that STAT3 KD in BLCA cells expressing normal SMARCB1 levels did not affect spheroid formation (Supplementary Fig. 11B). Conversely, STAT3 KD in SMARCB1 KO cells showed decreased spheroid formation (Supplementary Fig. 11C).

## STAT3 inhibition is an effective therapeutic strategy in SMARCB1 KO BLCA tumors

To investigate the therapeutic effect of STAT3 inhibition, we utilized the specific STAT3 inhibitor, TTI-101 which directly and competitively targets the Tyrosine phosphorylation (Y705) on STAT3[26]. We orthotopically engrafted SMARCB1 KO (clone C16) T24 BLCA cells. After 7

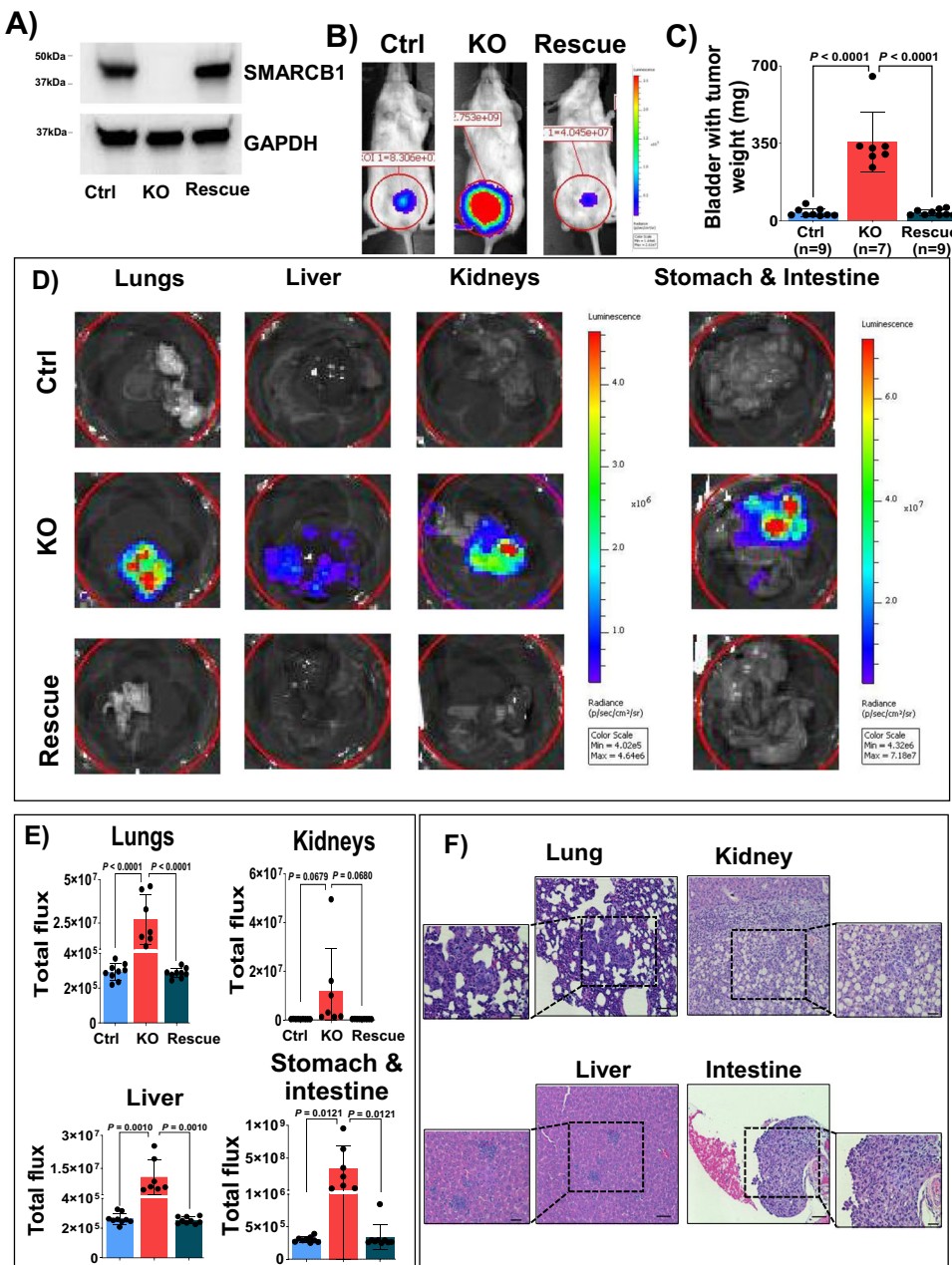

**Fig. 2 | Effect of SMARCB1 on BLCA tumor growth and metastasis. A** Validation of CRISPR/Cas9 based SMARCB1 KO (clone 16) and rescue (ectopic overexpression of SMARCB1 in KO derived from clone 16) by immunoblot analysis in the T24 BLCA cell line. GAPDH was used as loading control. **B** Representative bioluminescence images (BLI) of mice bearing T24 Control (ctrl), SMARCB1 KO, and SMARCB1 rescue orthotopic xenografts on day 15. **C** Weight of orthotopic mice bladders harboring tumors (endpoint, day 23) from T24 ctrl ($n = 9$), SMARCB1 KO ($n = 7$), and SMARCB1 rescue ($n = 9$) [Data are represented as mean ± standard deviation (SD)]. **D** Representative ex-vivo BLI images of metastatic lesions (lungs, liver, kidneys, and stomach & intestine) from mice bearing T24 ctrl, SMARCB1 KO, and rescue orthotopic xenografts. [Note: Representative ex-vivo BLI images were cropped from different non overlapping regions of same images for lungs, liver, kidneys, and stomach & intestine]. **E** Scatter plots represent the quantification of BLI signal of lungs, liver, kidneys, and stomach & intestine of mice bearing orthotopic xenografts from T24 ctrl ($n = 9$), SMARCB1 KO ($n = 7$) and rescue ($n = 9$) (quantified by BLI signal; photons/sec/cm²/sr) [Data are represented as mean ± standard deviation (SD)]. **F** Histology images of Hematoxylin and eosin (H&E) staining of ex-vivo metastatic organs derived from SMARCB1 KO metastatic lesions derived from panel (**D**). The metastatic lesions were highlighted with dotted boxes. Scale bar represents 100 μm. For panels **C** and **E**, P-values were determined by unpaired two-tailed Student's t-test. Source data are provided as a Source Data file.

days of implantation (Fig. 5A), mice were randomized to TTI-101 treatment or vehicle control based on BLI signal. Treatment of SMARCB1 KO (clone C16) xenografts with TTI-101 showed a significant decrease in BLI signal compared to the vehicle control (Fig. 5B), indicative of decreased tumor growth. Following 4 weeks of treatment (29 days; at endpoint), we observed a significantly decreased orthotopic bladder tumor weight in mice that were treated with TTI-101 as compared to vehicle treated mice (Fig. 5C). Furthermore, there was significantly decreased BLI signal at metastatic sites following TTI-101 treatment compared to vehicle control (Fig. 5D), suggesting decreased metastatic lesions. Moreover, we measured the plasma IL6 levels in TTI-101 and vehicle treated mice and observed a significant decrease of IL6 levels following TTI-101 treatment (Fig. 5E). Conversely, TTI-101 did not affect tumor growth, metastasis, or plasma IL6 levels compared

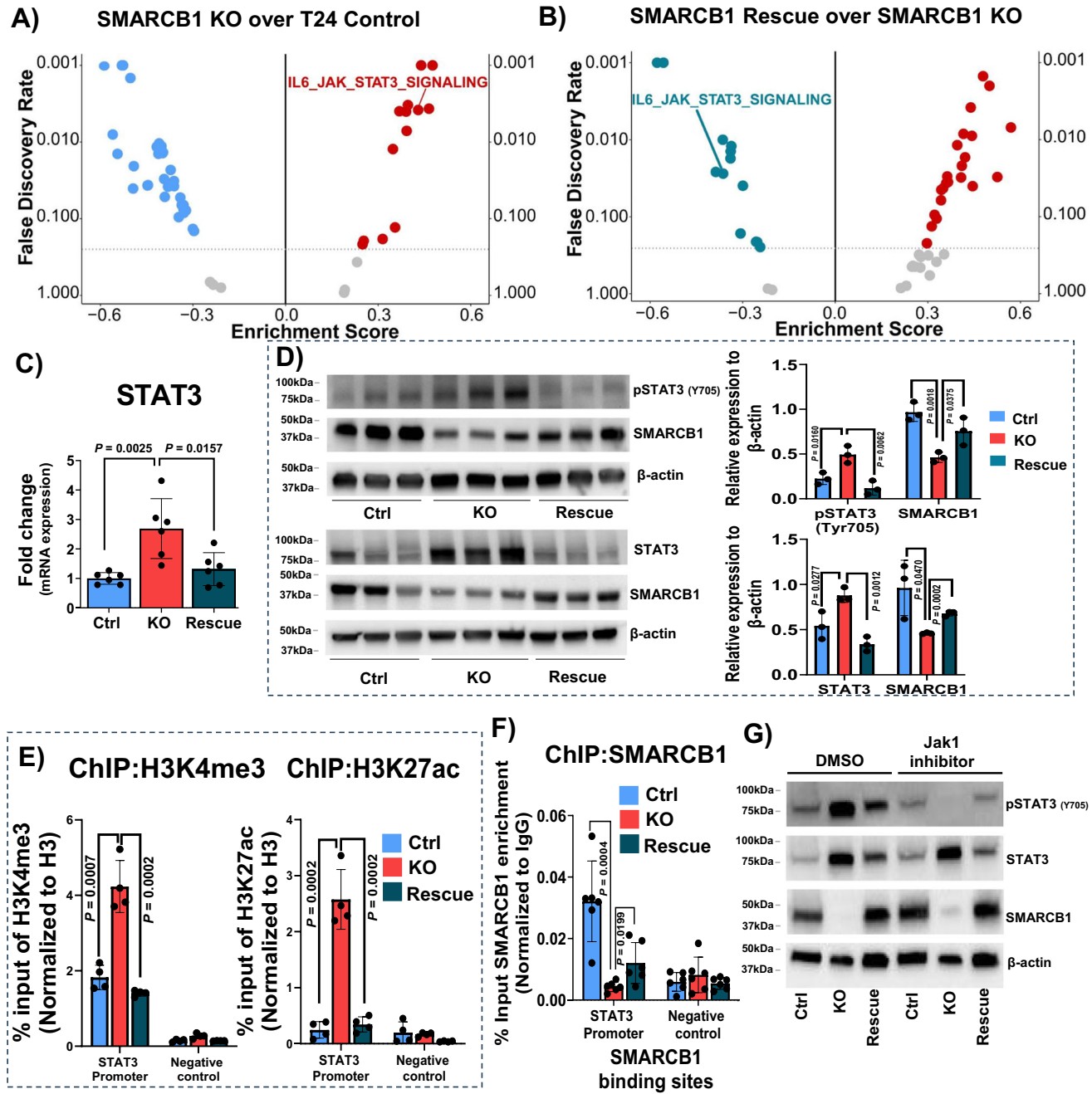

**Fig. 3 | SMARCB1 loss upregulates STAT3 expression. A**, **B** Gene set enrichment analysis (GSEA) of RNA-seq data from orthotopic tumors derived from T24 control, SMARCB1 KO and SMARCB1 rescue xenografts. Volcano plot represents the hallmark pathways that were enriched in **A** SMARCB1 KO over T24 control (FDR < 0.25) and **B** SMARCB1 rescue over SMARCB1 KO (FDR < 0.25). **C** Relative fold change in mRNA levels of STAT3 from T24 ctrl (*n* = 6 replicates sampled across three different mice; each one analyzed under three technical replicates), SMARCB1 KO (*n* = 6 replicates sampled across three different mice; each one analyzed under three technical replicates) and SMARCB1 rescue (*n* = 6 replicates sampled across three different mice; each one analyzed under three technical replicates) orthotopic xenografts. Normalized with β-actin [Data are represented as mean ± standard deviation (SD)]. **D** Immunoblot analysis of pSTAT3 (Y705), STAT3, SMARCB1 in T24 ctrl, SMARCB1 KO and SMARCB1 rescue orthotopic tumors (*n* = 3). Same lysate was used for upper and bottom panels. β-actin was used as loading control. Scatter

plots show relative expression of the target proteins after background subtraction and normalization to β-actin [Data are represented as mean ± standard deviation (SD)]. **E** ChIP-qPCR analysis shows increased levels of H3K4me3 and H3K27ac on STAT3 promoter in SMARCB1 KO xenografts (*n* = 4; two biological replicates analyzed under two technical replicates) [Data are represented as mean ± standard deviation (SD)]. **F** ChIP-qPCR analysis shows decreased levels of SMARCB1 on STAT3 promoter in SMARCB1 KO xenografts which was rescued in SMARCB1 re-expression (*n* = 6; three biological replicates analyzed under two technical replicates) [Data are represented as mean ± standard deviation (SD)]. **G** Immunoblot analysis of pSTAT3 (Y705), STAT3, and SMARCB1 in T24 ctrl, KO, and rescue spheroids treated with the JAK1 inhibitor, Itacitinib for 10 days at 1000 nM concentration. β-actin was used as loading control. For panels **C**, **D**, **E** and **F**, *P*-values were determined by unpaired two-tailed Student's *t*-test. Source data are provided as a Source Data file.

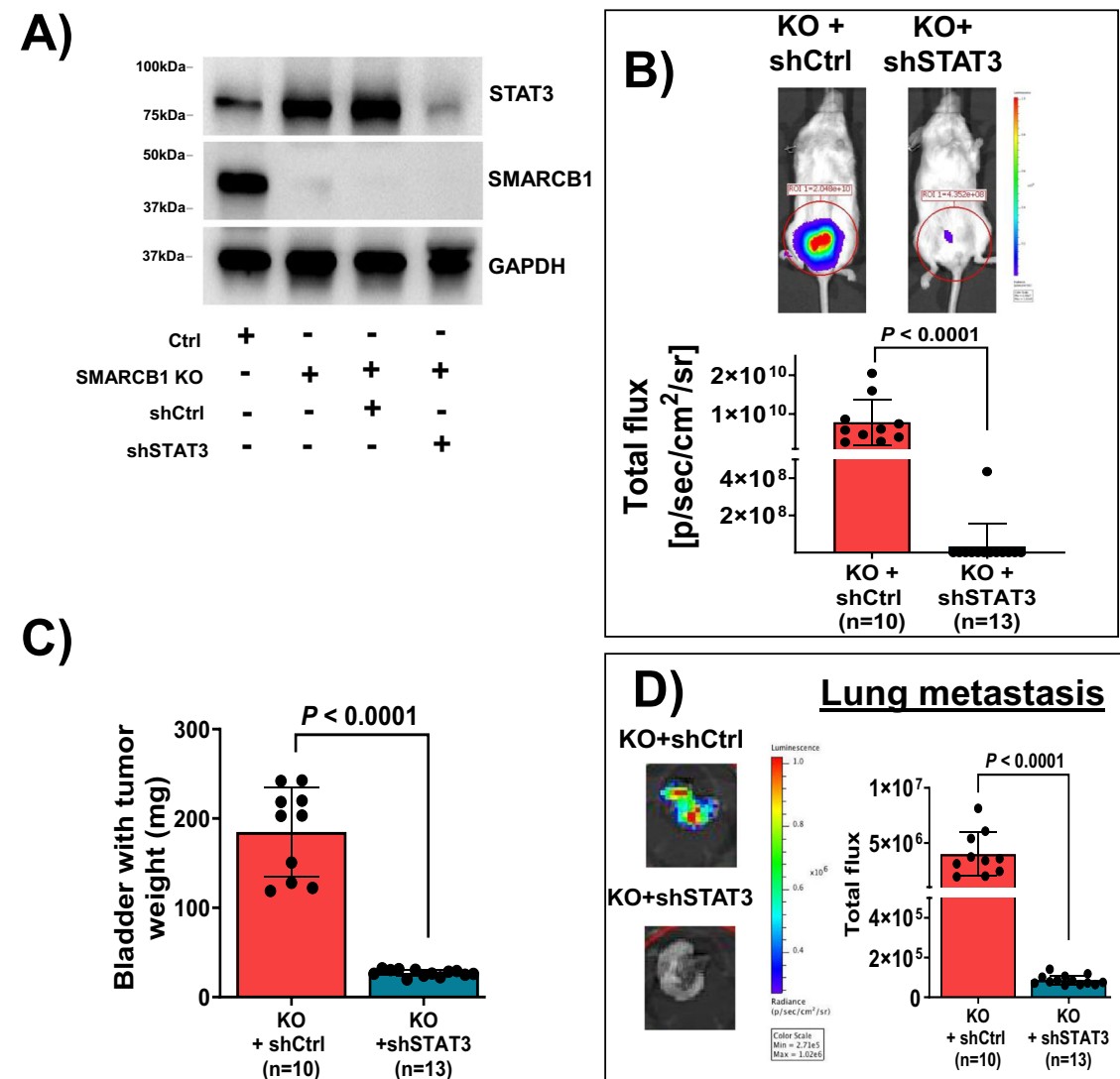

**Fig. 4 | STAT3 is essential for tumor growth and metastasis in SMARCB1 KO BLCA. A** Immunoblot analysis showing the confirmation of STAT3 knockdown (shSTAT3) in T24 SMARCB1 KO cell lines (clone 16). Lane 1- T24 ctrl; lane 2- SMARCB1 KO; lane 3- KO with shCtrl; and lane 4- KO with shSTAT3. Lane 3 and 4 cell lines were used for the in vivo experiments. GAPDH was used as loading control. **B** Top panel shows representative BLI in orthotopic mouse xenograft models showing decreased BLI signal indicating decreased tumor growth in SMARCB1 KO with shSTAT3. Bottom panel shows the scatter plot representing BLI signal on day 25 from xenografts derived from SMARCB1 KO with shCtrl ($n = 10$) and SMARCB1 KO with shSTAT3 ($n = 13$) [Data are represented as mean ± standard deviation (SD)]. **C** Scatter plot represents the weight of orthotopic mouse bladders harboring tumors (endpoint; day 27) from SMARCB1 KO + shCtrl ($n = 10$) and SMARCB1 KO + shSTAT3 ($n = 13$) [Data are represented as mean ± standard deviation (SD)]. **D** Left panel shows the representative ex-vivo BLI signal of lungs from mice bearing T24 SMARCB1 KO with scrambled sh or STAT3 KD in SMARCB1 KO orthotopic xenografts. Right panel shows the scatter plot representing the quantification of BLI signal of lungs from SMARCB1 KO with shCtrl ($n = 10$) and SMARCB1 KO with shSTAT3 ($n = 13$) [Data are represented as mean ± standard deviation (SD)]. For panels **B**, **C** and **D**, $P$-values were determined by unpaired two-tailed Student's $t$-test. Source data are provided as a Source Data file.

with vehicle control in orthotopic xenografts using T24 control cells (Supplementary Fig. 12A–C).

**TTI-101 inhibits tumor growth in SMARCB1-deficient BLCA PDXs**

We preclinically investigated the efficacy of TTI-101 in PDX models. The TM00020 PDX with SMARCB1 deletion was obtained from Jackson Labs (Supplementary Data 9). We further verified SMARCB1 mRNA expression by qRT-PCR and pSTAT3 (Y705) by IHC comparing with BCM-BL8091 PDX obtained from Baylor College of Medicine (BCM) Patient-Derived Xenograft (PDX) core. Our analysis revealed that TM00020 PDX had low SMARCB1 mRNA and high pSTAT3 (Y705) levels (Supplementary Fig. 13A–C). Based on the results we specified TM00020 as SMARCB1-deficient compared with BCM-BL8091 PDX (high SMARCB1). Mice harboring the SMARCB1-deficient

PDX were treated daily with TTI-101 or vehicle control for 14 days whereas those harboring the SMARCB1 high PDX were treated daily for 77 days. Similarly, to the orthotopic SMARCB1 KO xenografts, the SMARCB1-deficient PDX model demonstrated a significant inhibition of tumor growth with TTI-101 compared to vehicle treatment (Fig. 6A, B). However, TTI-101 treatment did not significantly affect the tumor growth in BCM-BL8091 (high SMARCB1) BLCA PDX model (Fig. 6C, D) even after 77 days of treatment. Total body weight of the mice was monitored during treatment, and we observed no significant loss of body weight with TTI-101 treatment in either PDX model (Supplementary Fig. 13D–E). Collectively, these results provide a strong pre-clinical rationale for TTI-101 as a therapeutic strategy for SMARCB1-deficient BLCA with concomitant activation of pSTAT3 (Fig. 6E).

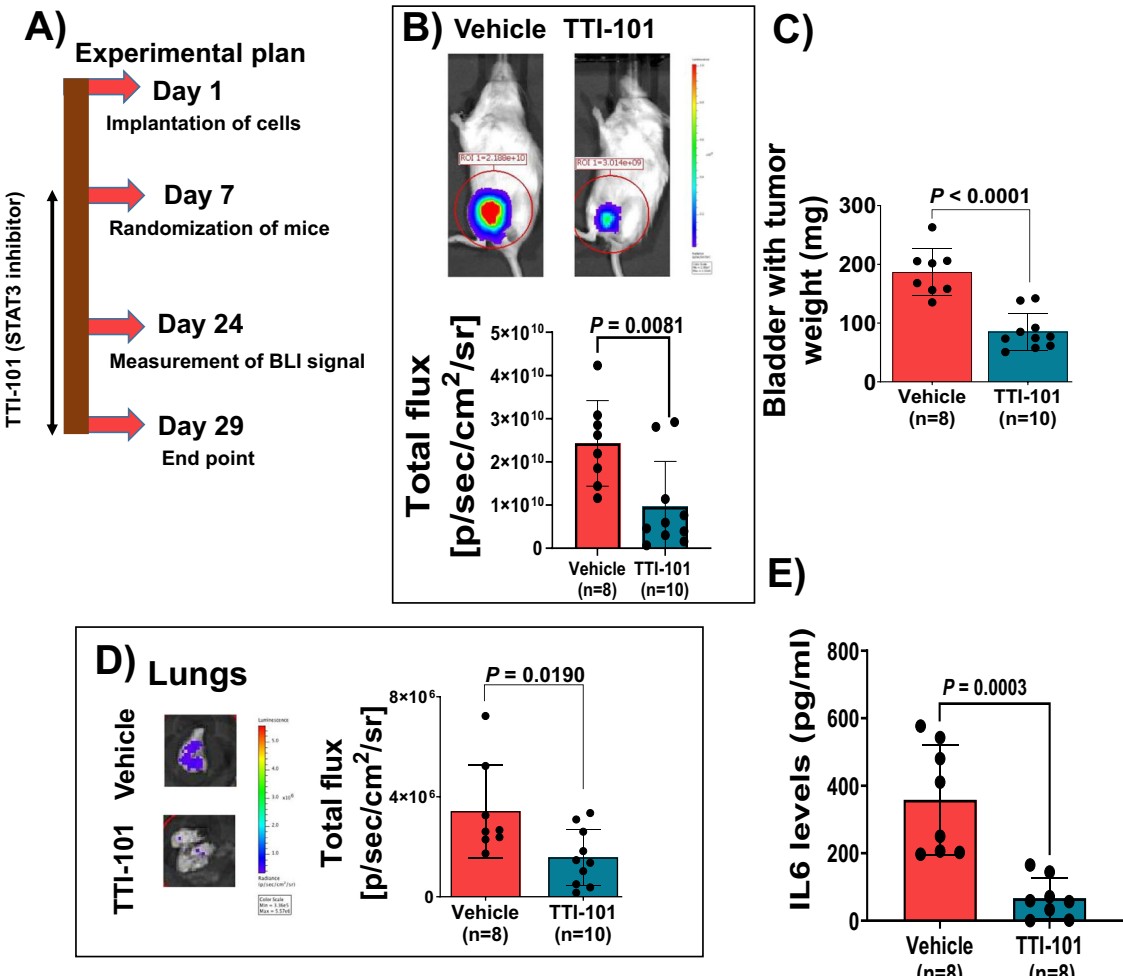

**Fig. 5 | Therapeutic targeting of pSTAT3 (Y705) in SMARCB1 KO orthotopic BLCA xenografts. A** Schematic overview of experimental plan for SMARCB1 KO xenografts with STAT3 inhibitor TTI-101. **B** Top panel shows representative BLI images of mice bearing SMARCB1 KO orthotopic xenografts which were treated with vehicle ($n = 8$) and TTI-101 ($n = 10$; on day 24). Bottom panel shows scatter plot representing BLI signal on day 24 from vehicle and TTI-101 treated mice (oral gavage – 50 mg/kg body weight, twice a day) [Data are represented as mean ± standard deviation (SD)]. **C** Scatter plot showing weight of orthotopic bladders harboring tumors (day 29) from T24 SMARCB1 KO treated with vehicle ($n = 8$) and TTI-101 ($n = 10$) [Data are represented as mean ± standard deviation (SD)]. **D** Left

panel shows representative ex-vivo BLI images of lungs of T24 SMARCB1 KO cell-bearing mice treated with vehicle ($n = 8$) and TTI-101 ($n = 10$) at endpoint day 29. Right panel shows scatter plot representing the quantification of BLI signal of lungs [Data are represented as mean ± standard deviation (SD)]. **E** Quantification of IL6 secretion by ELISA from mouse plasma bearing T24 SMARCB1 KO xenografts treated with vehicle ($n = 8$ mice) and TTI-101 ($n = 8$ mice) inhibitor at endpoint day 29 [Data are represented as mean ± standard deviation (SD)]. For panels **B**, **C**, **D**, and **E**, *P*-values were determined by unpaired two-tailed Student's *t-test*. Source data are provided as a Source Data file.

## Identification of transcriptional signature from SMARCB1 loss xenografts to predict SMARCB1-deficient BLCA

We identified 393 genes (59 upregulated and 334 downregulated) that are deregulated upon SMARCB1 loss and restored upon gain of SMARCB1 in T24 BLCA xenografts (clone 16) (Supplementary Fig. 14A–F; Supplementary Data 10). We then examined the ability of 55 out of 59 upregulated genes to distinguish patients with SMARCB1-deficient BLCA (Refer to Fig. 1E). To assess the discrimination performance, using the logistic regression model as a classifier and as shown in Supplementary Fig. 14G, the concordance probability between predicted (using the 59 gene signature) and observed SMARCB1-deficient BLCA versus all other BLCA was 76%.

## Discussion

As many as 50% of patients with muscle invasive BLCA may have occult metastasis that becomes clinically apparent within 5 years of initial diagnosis, and around 5% will have distant metastasis at the time of initial diagnosis. The lymph nodes, bones, lungs, liver, and peritoneum

are the most common sites of metastasis in BLCA. In addition to traditional chemotherapy[27], a variety of new therapeutic agents including immune checkpoint therapy, FGFR3 inhibitor, and antibody-drug conjugates have been approved by the FDA for the treatment of metastatic BLCA as recently reviewed[28]. However, about 30-80% of bladder cancer patients still do not respond to these therapeutic agents. Thus, identifying novel therapeutic targets that drive metastasis and poor patient outcomes is of paramount importance for the development of biomarker-guided, genomic alteration-based, effective therapies for patients with metastatic BLCA.

In this study, we demonstrate that SMARCB1 deficiency, defined as the presence of deep or shallow SMARCB1 deletion and low SMARCB1 mRNA, may drive BLCA disease progression in approximately 32% of patients with BLCA. These findings are in line with existing reports suggesting that SMARCB1 loss is a frequent driver of tumorigenesis across multiple malignancies[10,12,15,17,29–32]. This relationship is predominantly due to somatic, as opposed to germline, SMARCB1 alterations. An analysis of pathogenic germline variants in

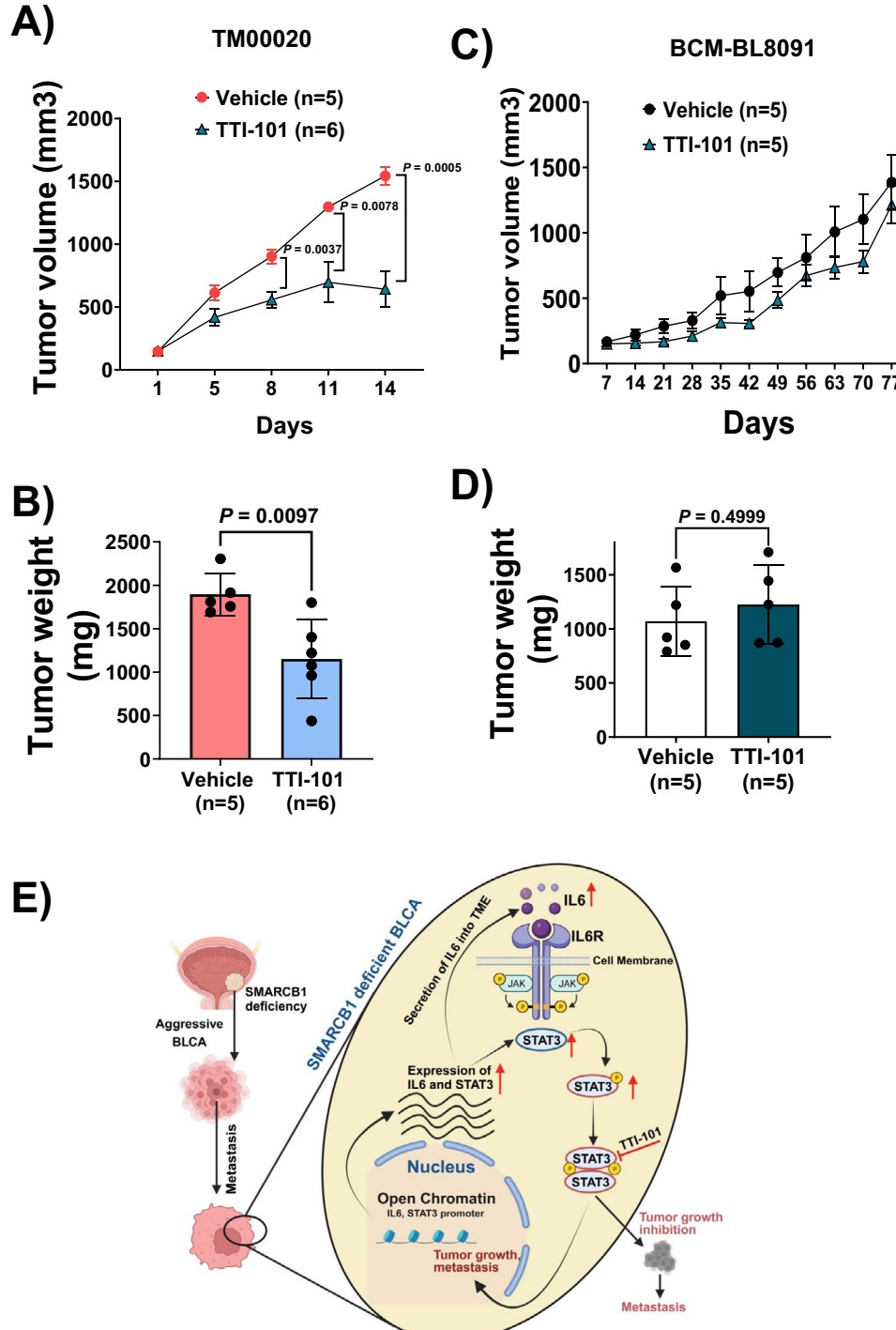

**Fig. 6 | STAT3 inhibitor TTI-101 suppresses tumor growth in SMARCB1-deficient patient-derived xenografts. A** Tumor growth analysis of PDX model [TM00020 obtained from The Jackson laboratory; harboring SMARCB1 deletion (Supplementary Data 9) and low mRNA (Supplementary Fig. 13A) treated with vehicle ($n = 5$) and TTI-101 ($n = 6$) for a period of 14 days at 50 mg/kg by oral gavage, twice per day]. [Data are represented as mean ± standard error of mean (SEM); unpaired two-tailed Student's *t-test*]. **B** Weight of tumors (after 14 days) from TM00020 PDX treated with vehicle ($n = 5$) and TTI-101 ($n = 6$). [Data are represented as mean ± standard deviation (SD); unpaired two-tailed Student's *t-test*]. **C** Tumor growth analysis of PDX [BCM-BL8091 obtained from BCM PDX core; high SMARCB1 mRNA (Supplementary Fig. 13A)] treated with vehicle ($n = 5$) and TTI-101 ($n = 5$) inhibitor for a period of 77 days at 50 mg/kg by oral gavage, twice per day. [Data are represented as mean ± standard error of mean (SEM)]. No significant difference was observed at all time points using unpaired two-tailed Student's *t-test*. **D** Weight of tumors (after 77 days) from BCM-BL8091 PDX treated with vehicle ($n = 5$) and TTI-101 ($n = 5$). [Data are represented as mean ± standard deviation (SD); unpaired two-tailed Student's *t-test*]. **E** Schematic representation of the finding between SMARCB1 deficiency and the IL6/JAK/STAT3 signaling axis in BLCA. TME – Tumor microenvironment. Source data are provided as a Source Data file.

10,389 cancers[33] identified one patient with ovarian cancer and a SMARCB1 mutation. In addition, up to 20% of malignant rhabdoid tumors have germline SMARCB1 deletions or mutations[14]. However, no association between SMARCB1 germline mutations and BLCA has been reported to date. Future studies should investigate the possibility that polygenic germline interactions may influence SMARCB1 expression in urothelial carcinomas.

Mechanistically, we noted in vitro and in vivo that SMARCB1 deficiency may drive BLCA growth by engaging the IL6/JAK/STAT3 signaling pathway. ATAC-seq and ChIP-qPCR analyses revealed increased accessibility of the STAT3 promoter upon both complete (KO) and partial (KD) SMARCB1 loss. We subsequently investigated the efficacy of the pSTAT3 selective inhibitor, TTI-101, in SMARCB1 KO orthotopic BLCA cell line-derived xenografts as well as SMARCB1-deficient PDX model. TTI-101 durably inhibited in vivo tumor growth throughout the course of drug treatment in both the orthotopic SMARCB1 KO cell line-derived xenograft and the SMARCB1-deficient PDX model. Furthermore, TTI-101 potently inhibited metastatic spread in the orthotopic xenograft model. Conversely, TTI-101 was ineffective in SMARCB1 high orthotopic cell line-derived xenografts and PDX model.

Our data demonstrate the preclinical efficacy of TTI-101 in SMARCB1-deficient tumors and suggest that SMARCB1 deficiency may predict response to STAT3 inhibition in metastatic BLCA. TTI-101 is being tested across solid tumors in a phase I study (NCT03195699) and SMARCB1 deficiency can be considered as a predictive biomarker for evaluating TTI-101 clinical efficacy. Based on our preclinical findings, we propose that TTI-101 merits clinical investigation in the subset of patients with SMARCB1-deficient BLCA. Towards this purpose, we have also identified a gene signature to identify SMARCB1 deficiency in BLCA and accordingly help screen for patients who could benefit from STAT3 therapy. Further research is warranted to elucidate the impact of other pathways beyond IL6/JAK/STAT3 in SMARCB1-deficient BLCA.

## Methods
All the experiments for the study were performed following standards according to the protocol approved by the Institutional Biological & Chemical (IBC) Safety Committee, Baylor College of Medicine (BCM) under the protocol number D-499. All mouse experiments were performed at BCM and approved following the guidelines of BCM Institutional Animal Care and Use Committee (IACUC) under the protocol number AN-7324. BCM-BL8091 patient-derived xenograft (PDX) was acquired from the BCM PDX core. The PDX core obtained approval for the generation of the PDX through IRB protocol H-25708 and subsequently implanted it under the IACUC-approved protocol number AN-2289. All the BLCA patients' tissues specimens (PDXs and FFPE) used in this study were obtained by informed patient consent.

### Kaplan–Meier analysis and GSEA analysis of TCGA-BLCA
SMARCB1 mRNA expression and survival data of TCGA-BLCA (Refer to Fig. 1A) dataset was downloaded from the National Cancer Institute Genomic Data Commons (NCI-GDC) Data Portal[34] using the R package TCGA biolinks (v 2.12.6)[35], and SMARCB1 copy number alterations of TCGA-BLCA[7] (Refer to Fig. 1D; Supplementary Fig. 2E) dataset was downloaded from cBioPortal.

For the preprocessing of survival data, the overall survival day is defined as the maximum value in days to last follow-up and days to death. Additionally, patients with missing overall survival day or vital status were removed. Patients with bladder cancer were divided into SMARCB1 high group and SMARCB1 low group based on their SMARCB1 gene expression values using maximally selected rank statistics[18] (Refer to Source data). The survival analysis was conducted using the R package Survival (v 3.2-11) and the R package Survminer (v 0.4.9) for the generation of figures.

Gene set enrichment analysis (Refer to Fig. 1B, C) of TCGA-BLCA was performed by R package fgsea (version 1.14.0)[36] and plotted by ggplot2 (version 3.3.5)[37]. By GSEA method[38], enrichment of genes in the HALLMARK IL6/JAK/STAT3 gene set, within bladder tumors with low SMARCB1 expression which was defined based on maximally selected rank statistics.

We delved into the GSEA focusing on SMARCB1 copy number alterations from TCGA-BLCA (Supplementary Fig. 2D). Specifically, we compared gene expression between SMARCB1 shallow/deep deletion vs SMARCB1 diploid. Differential gene expression analysis was carried out using DESeq2 (v 1.28.1)[39], and the subsequent GSEA was conducted using the GSEA software[40].

### Correlation of SMARCB1 copy number alterations, mRNA expression, and pSTAT3 protein expression from TCGA-BLCA
As described above, copy number alterations, mRNA for SMARCB1 was obtained from TCGA-BLCA were downloaded from the NCI-GDC Data Portal[34,40,41] by the R package and pSTAT3 from reverse-phase protein array (RPPA) from the broad institute's firehose data portal (http://gdac.broadinstitute.org).

Next, we selected the matched patients from (1) TCGA-BLCA contains the pSTAT3 protein from RPPA, (2) SMARCB1 copy number alterations from TCGA-BLCA[7], and (3) SMARCB1 mRNA (maximally selected rank statistics) from TCGA-BLCA[34] used for KM plot to divide the patient population into following six groups. Group I: SMARCB1 shallow/deep deletion with low SMARCB1 mRNA, Group II: SMARCB1 diploid with low SMARCB1 mRNA, Group III: SMARCB1 gain with low SMARCB1 mRNA, Group IV: SMARCB1 shallow deletion with high SMARCB1 mRNA, Group V: SMARCB1 diploid with high SMARCB1 mRNA and Group VI: SMARCB1 gain/amplification with high SMARCB1 mRNA. Data was analyzed using GraphPad Prism v10.

### GSEA analysis of publicly available BLCA cohorts (GSE48276, GSE32548 and GSE31684)
For GSEA analysis for Supplementary Fig. 2A, B, C, genes in each dataset[20–22] were ranked from low to high Pearson's correlation with SMARCB1 expression (using log2- transformed expression values). Gene Set Enrichment Analysis (GSEA) was performed using the GSEA software[38].

### SMARCB1 mRNA and copy number analysis of pan-cancer from TCGA
For SMARCB1 copy number analysis in TCGA pan-cancer (BLCA was excluded in this analysis), the copy number alterations were downloaded from the broad institute's firehose data portal (http://gdac.broadinstitute.org). The copy number alterations were mapped to corresponding SMARCB1 mRNA expression levels and represented as a violin plot using GraphPad Prism v10.

### Generation of SMARCB1 KO cells from human bladder cancer T24 cell lines
Knockout of SMARCB1 was achieved by lentivirus generated in HEK-293T cells using pLentiCRISPR v2 plasmids harboring gRNA (guide RNA). The generation and infection of T24 bladder cancer cell lines were done according to a previously published study[17]. Puromycin selection was performed to select lentiviral transduced single-cell clones. As a negative control, we used cells transduced by lentivirus generated using pLentiCRISPR v2 plasmid harboring non-targeting control gRNA and selected with puromycin (Sigma, MO). The sequences for generating the SMARCB1 knockout and non-targeting control gRNA were provided in Supplementary Table 1. The cell lines were tagged with luciferase by blasticidin selection for tracking the metastatic potential and tumor growth.

## DNA isolation and Sanger sequencing of PCR products

DNA was isolated from cell pellets using a QIAGEN DNA isolation kit (QIAGEN). PCR was performed with primers (Supplementary Table 2) flanking the targeted genomic region. PCR products were then run on 1% agarose gels and purified with a QIAGEN Gel extraction kit (QIA-GEN). 10 ng of purified PCR product was then verified by Sanger sequencing and the obtained peaks were analyzed by Chromas software and shown in Supplementary Fig. 3.

## Generation of orthotopic cell line-derived xenografts

For orthotopic xenografts, cells were resuspended in 25 μl of media. The resuspended cells were then mixed with 25 μl of Matrigel (Corning, NY) for the engraftment. After mice randomization, ~ 50 μl of Matrigel-media mix was then orthotopically injected into the bladder wall of 6–8-week-old NOD/SCID/IL2rγnull (NSG) (Jackson labs, USA) female mice. To measure the tumor growth and metastasis, a total of 200,000 control or SMARCB1 KO (C16 or C45) or rescue cells (Fig. 2B–D and Supplementary Fig. 4A and 4C) or SMARCB1 KD cells (Left panel of Supplementary Fig. 4G) were injected into the bladder wall. 100,000 cells of SMARCB1 KO with shCtrl or STAT3 KD (Fig. 4) were injected into the bladder wall. T24 SMARCB1 KO therapeutic experiments 50,000 cells (Fig. 5) were injected into the bladder wall. T24 control cells for therapeutic experiments, a total of 500,000 cells (Supplementary Fig. 12), were injected into the bladder wall. After seven days of the implantation of cells for orthotopic in vivo experiments, wound clips were removed, and the bioluminescence imaging (BLI) signal was measured. The measurement of BLI signal was performed by injecting 100 μl of luciferin (15 mg/ml in PBS; GoldBio, USA) substrate via the retro-orbital route in anesthetized mice. The weight of the mice was measured and recorded weekly. BLI signal measurements were plotted using GraphPad Prism v10.

## Ex-vivo BLI measurements from organs

The body weight of mice was used to determine the time for euthanasia. Before euthanizing the mice, 100 μl of luciferin was retro-orbitally injected and euthanasia was performed by cervical dislocation under anesthetic conditions. Bladder, lungs, liver, kidneys, stomach, and intestine organs were immediately collected, and imaged for BLI signal using an In Vivo Imaging System (IVIS). The measured BLI signal was quantified using IVIS lumina software (Perkin Elmer, MA) and plotted using GraphPad Prism v10. Raw data was available in Source Data.

## RNA isolation from mouse orthotopic xenografts and PDX tumors

For transcriptomics analysis from mouse xenografts, RNA was isolated from frozen tissues using the QIAGEN RNAeasy microkit (QIAGEN, Germany). Genomic DNA digestion was performed to remove traces of genomic DNA during RNA extraction. RNA quantification from frozen tissues was measured by Qubit and the RNA quality was determined using Agilent Tapestation 4200 (Agilent Technologies, CA).

## RNA sequencing from mouse orthotopic xenograft tumors

For RNA sequencing from frozen tissues derived from mouse xenografts, 100 ng of total RNA was used for preparing sequencing libraries using RNA HyperPrep Kit with RiboErase (HMR) kit (Roche sequencing solutions). The first depletes rRNA from total RNA. After depletion of ribosomal RNA, the remaining RNA is digested by DNaseI (Thermo-Scientific, MA) to remove traces of genomic DNA contamination. For cDNA synthesis, RNA was purified, fragmented, and primed. It is then reverse transcribed into first-strand cDNA using random primers. The next step is the removal of the RNA template and synthesis of a replacement strand, incorporating dUTP in place of dTTP to generate double-stranded cDNA. To separate double-stranded cDNA from the second strand reaction mix pure beads (KAPA BIOSYSTEMS) are used.

This resulted in blunt-ended cDNA. 3′ends of blunt fragments were added with a single 'A' nucleotide. Next, multiple indexing adapters, containing a single 'T' nucleotide on the 3′end of the adapter, are then ligated to the ends of double-stranded cDNA preparing them for hybridization onto flow cells. These adapter libraries were then amplified by PCR, purified using pure beads, and validated for appropriate size on a 4200-tapestation D1000 screen tape (Agilent Technologies, Inc.). Quantification of DNA libraries was performed using the KAPA Biosystems qPCR kit and was pooled together in an equimolar fashion. Each pool of DNA library was denatured and diluted to 350pM with 1% PhiX control library. This was then loaded onto a NovaSeq reagent cartridge for 100 paired-end sequencing and sequenced on a NovaSeq6000 according to the manufacturer's instructions (Illumina Inc.).

## SMARCB1 mRNA expression from patient tumors

For Supplementary Fig. 7B, the plotted data originates from RNA sequencing performed in Dr. Apolo Andrea's lab at the National Cancer Institute, under the IRB approved study number 16-C-0121 and shared the processed data for this study was used to check SMARCB1 expression.

Briefly, for transcriptomics from FFPE tissues of matched primary and metastatic patient samples, total RNA isolation protocol from Illumina (San Diego, CA) was used. This protocol involves the removal of ribosomal RNA (rRNA) using biotinylated, target-specific oligos combined with Ribo-Zero rRNA removal beads (Illumina, CA). First-strand cDNA was synthesized by reverse transcribing with reverse transcriptase and random primers, followed by synthesis of second strand using DNA polymerase I and RNase H (Roche, USA). The synthesized double-strand cDNA is used as input for Standard Illumina library prep with end-repair, ligation with adapters, followed by PCR amplification that would go into the sequencer. Quantification of the final purified product was done by qPCR before cluster generation and sequencing.

For RNA sequencing from FFPE tissues (Dr. Apolo Andrea's lab, NCI), the TruSeq Stranded RNA libraries were pooled and sequenced on one NovaSeq6000 (Illumina, CA) using 2 × 151 cycles kit for pair-end run. For processing raw data files, the HiSeq Real-Time Analysis software (RTA 3.4.4) was used. Demultiplexing and conversion of binary base calls and qualities to fast q formation were done using the Illumina bcl2fastq2. Sequencing reads were trimmed, and low-quality bases were removed using Cutadapt (version 1.18). The trimmed reads were mapped to mouse reference genome mm10 and GENCODE annotation v21 using STAR aligner (version 2.7.0 f) with a two-pass alignment option. RSEM (version 1.3.1) was used for gene and transcript quantification based on the GENCODE annotation file. All of these experiments were carried out at Dr. Apolo Andrea's lab at NCI, and the processed data was shared to check the SMARCB1 expression.

## RNA-seq data processing from mouse xenografts and statistical analysis

We performed RNA-seq analysis comparing control vs. SMARCB1 KO and SMARCB1 KO vs. rescue to identify differentially expressed genes. To mitigate the effect of mouse mRNA contamination and estimate mouse contamination rates, we used both competitive and non-competitive quantification methods described below.

Quality control and adapter trimming were conducted on the raw sequencing reads (FASTQ files) using Trim-galore (v 0.6.6). We then concatenated the human reference transcripts (GENCODE GRCh38) and the mouse reference transcripts (GENCODE GRCm39)[42] to perform the first competitive quantification analysis using Salmon (v 1.4.0)[43]. The concatenated reference transcripts were used to evaluate the transcript expression levels from trimmed sequencing reads. The

contamination rates estimated by Salmon were calculated using Eq. (1). The estimated human transcript-level expression was further summarized to gene-level expression using the R package Tximport (v 1.16.1)[44].

$$Salmon\_Estimation = (Sum(count\ of\ mouse\ transcripts))/$$
$$(Sum(count\ of\ mouse\ transcripts) + Sum(count\ of\ human\ transcripts))$$
$$(1)$$

We used STAR (v 2.7.8a)[45] for the second competitive quantification method to obtain a comprehensive view of mouse contamination. To accomplish this, we first annotated the genome transcripts with chromosome labels for the human (GENCODE GRCh38) and the mouse reference genome (GENCODE GRCm39). Then, these annotated reference genomes from human and mouse were concatenated together. The same annotation and file combination processes were performed for the human (GENCODE GRCh38) and the mouse gene annotation file (GENCODE GRCm39). The trimmed sequencing reads from different samples were then aligned to the combined human and mouse reference genome. The contamination rates estimated by STAR can be calculated using Eq. (2).

$$STAR\_Estimation = (Sum(number\ of\ mouse\ reads))/$$
$$(Sum(number\ of\ mouse\ reads) + Sum(number\ of\ human\ reads))$$
$$(2)$$

For the non-competitive quantification method, we first used STAR to align trimmed sequencing reads to both the human reference genome (GENCODE GRCh38) and the mouse reference genome (GENCODE GRCm39). The mouse reads were then filtered using the R package XenofilteR (v 1.6)[46] which also generated an estimation of contamination rates (Supplementary Data 6C). The raw read counts were generated by HTSeq-count (v 0.12.4)[47].

After the quantification by either competitive or non-competitive method, we used the R package DESeq2 (v 1.28.1)[39] to normalize the raw count and conduct differential analysis of gene expression. Mouse contamination rates estimated by both Salmon/XenofilteR were added to the expression model of DESeq2 as a covariate to improve the accuracy of detecting differentially expressed genes. The differentially expressed genes (FDR < 0.05, Fold Change >2) identified by both the competitive and non-competitive methods were reported, used for further analysis, and represented as heat maps (Supplementary Fig. 14A–D).

### GSEA analysis of transcriptomics from mouse xenografts
GSEA was performed using the GSEA software[38]. The gene list that ranked based on the test statistics from DESeq2 served as an input to the GSEA pre-rank test. Additionally, the correlation analysis on normalized enrichment scores (NES) was performed to test the result of the GSEA. The hallmark gene sets that were significant (FDR < 0.25) under both the competitive and the non-competitive methods were reported and represented as volcano plots. GSEA analysis was performed to determine a significant p-value and p-values were determined using the Kolmogorov–Smirnov test and permutation test (One-sided and Benjamini–Hochberg corrections).

### Immunohistochemistry (IHC) and Hematoxylin and eosin (H&E) staining
IHC was performed on formalin-fixed paraffin-embedded tumor tissue sections. Five μM sections were used for the IHC analyses. The sections were stained with mouse monoclonal SMARCB1 antibody (BD Biosciences, USA), and mouse monoclonal pSTAT3 (Y705) antibody (Santha Cruz, USA), and the dilutions were specified in Supplementary Table 3. IHC staining was scored by a board-certified anatomic pathologist for extent and intensity. The extent score represents the estimated percentage of tumor cells with positive staining. All staining steps were performed using standard IHC protocols. The intensity score represents the average staining intensity of all positive cells in the section, such that a score of 0 is negative, 1 is weak, 2 is moderate, and 3 is strong. The total staining core or immunoreactive (IRS) score is the product of the intensity and extent scores. All the positive and negative controls for IHC were available in Source Data. All IHC stains were interpreted in conjunction with Hematoxylin and eosin-stained sections.

### RNA isolation and qRT-PCR analysis
RNA extraction from mouse xenografts was carried out as described above (RNA isolated from mouse xenografts). cDNA was synthesized from 1 μg of RNA using qScript cDNA Supermix (Quantabio). Primers for the target genes were obtained from IDT technologies, and their sequences are available upon request. Real-time PCR was performed using SYBR green master mix (Life Technologies, CA) or Taqman probes (Sigma, MO). The primer sequences used in this study are provided in Supplementary Table 4. mRNA expression was normalized using respective housekeeping genes (18srRNA or GAPDH or β-actin) and fold change was calculated by averaging technical replicates. The fold change values were analyzed using GraphPad Prism v10.

### Enzyme-linked immunosorbent assay (ELISA) in both in vitro and in vivo
BLCA cells were cultured in a conditioned medium (Media with 2% FBS) for 24 h. The conditioned medium was diluted at 1:50 with media (without FBS) and the IL6 levels were quantified using ELISA kits from BioLegend (San Diego, CA).

For ELISA from mouse plasma, blood was collected retro-orbitally in which the mice were penetrated with a capillary tube. Blood was drawn into Microvette® CB 300 Lithium heparin LH, capillary blood collection tube (Sarstedt AG & Co.KG) and spun down at 9600 × $g$ for 10 min at 4 °C to separate the red blood cells and plasma. Plasma was then diluted to 1:10 with the ELISA buffer and measured the IL6 levels in 96 well plate in duplicates.

For ELISA measurements from cell lines and mouse plasma, a standard curve was plotted with known IL6 concentrations, and an interpolation equation was used to calculate the final concentration of IL6 in conditioned media and plasma samples. Measurements of IL6 from plasma were run in duplicates and the average of IL6 concentration was taken for representation using GraphPad Prism v10. In the case of cell lines, the conditioned media was run in four replicates and an average of IL6 was represented.

### Spheroid assays and JAK1 inhibitor treatment
For spheroid assays, 5000 cells were resuspended in 25 μl of media. The suspension was then mixed with 25 μl of Matrigel and plated on non-adherent plates as a spheroid for one hour at 37 °C. After incubation, growth media was added to the cells and cultured. Brightfield images of the spheroids were captured at 2.5× magnification and then the different fields of view stitched together using Biotek Gen5 software. BLI measurements for the spheroids were taken by adding 100 μl of luciferin (15 mg/ml) to the wells and luminescence was captured by IVIS. All the experiments were performed three times independently. Spheroids from T24 Control, SMARCB1 KO, and rescue cell lines were treated with JAK1 inhibitor (Itacitinib) for 10 days at 1000 nM concentration. At the terminal point, spheroids were incubated with cell dissociation solution (Corning, USA) for 1 h on ice, and protein lysates were prepared using the RIPA buffer (Sigma, MO).

### Immunoblotting
Cell pellets and mouse xenograft tumors were homogenized using the RIPA buffer (Sigma, MO). After homogenization, the protein lysates were sonicated for 30 s and centrifuged at 18800 × $g$ for 30 min at 4 °C.

The supernatant was separated and transferred into a fresh tube. Protein concentration was measured using Pierce BCA protein assay kit (ThermoScientific, MA). Antibodies used for this study were listed in the Supplementary Table 3. GAPDH and β-actin were used as housekeeping genes. Multiple target proteins were detected on the same blot by Restore™ stripping buffer (ThermoScientific, MA). For the development of immunoblot, the western blot super signal Immobilon western chemiluminescent HRP substrate (Millipore, MA) was used. Immunoblots were quantified using Image J software by considering the integrating density and relative expression values for the target proteins were analyzed and normalized to associated housekeeping genes. All the raw images are available in the Source Data. All the experiments were performed three times independently.

### Reverse-phase protein array (RPPA) analysis
RPPA assays were performed as described previously with minimal changes[48–50]. Protein lysates were prepared from spheroids. Protein lysates at a concentration of 0.5 mg/ml were denatured in SDS sample buffer with 2.5% (vol/vol) 2-mercaptoethanol at 95 °C for 5 min, and then were spotted onto nitrocellulose-coated slides (Grace Bio-labs, Bend, OR) by an Quanterix 2470 Arrayer (Quanterix , Billerica, MA), with an array format of 960 lysates/slide (2880 spots/slide). Before primary antibody incubation, each slide was blocked with 1X I-block (Thermo Fisher, Waltham, MA). Each slide was stained with a targeted antibody, using an automated slide stainer, Autolink 48 (Agilent, Santa Clara, CA). One slide was incubated with antibody diluent instead of primary antibody and served as a negative control. Subsequently, the slides were incubated with a biotinylated secondary antibody followed by tyramide signal amplification (TSA) using the VECTASTAIN Elite ABC-HRP Kit (Vector Laboratories, Burlingame, CA) and TSA plus biotin (AKOYA Biosciences, Marlborough, MA). Visualization was completed with the addition of streptavidin-conjugated IRDye680 fluorophore (LI-COR Biosciences). There were 3 washes with 1× TBS-T (Dako, Carpinteria, CA) following incubation steps during the staining process. Total protein was assessed by fluorescent staining with Sypro Ruby Protein Blot Stain following the manufacturer's instructions (Molecular Probes, Eugene, OR).

The fluorescence-labeled slides were scanned on a GenePix 4400 AL scanner, and images were analyzed with GenePix Pro 7.0 software (Molecular Devices, Sunnyvale, CA). The fluorescence signal intensities of each spot were obtained after subtraction of the negative control signal and were then normalized against the total protein signal as described[48]. The normalized data were used for further analysis by taking the average of three technical replicates using GraphPad Prism v10.

### Library preparation for ATAC-seq using BLCA cell lines
A total of 50,000 cells per replicate were prepared for ATAC-seq libraries using the Omni-ATAC protocol[51]. To isolate nuclei, cells were resuspended in 50 µl of lysis buffer (0.1% Tween-20, 0.1% NP-40, and 0.01% digitonin in RSB buffer (10 mM Tris−HCl pH 7.5, 10 mM NaCl, 3 mM MgCl2)) and incubated on ice for 3 min. RSB buffer supplemented with 0.1% Tween-20 was used to wash out lysis buffer, and then nuclei were pelleted by centrifugation for 10 min at $500 \times g$. For transposition, nuclei were resuspended in 50 µl of Transposition mix plus 25 µl of Tagmentation DNA buffer (2.5 µl Tagment DNA enzyme (Illumina), 25 µl of Tagmentation DNA buffer (Illumina), 0.1% Tween-20 and 0.01% Digitonin) and incubated for 30 min at 37 °C. A MinElute PCR purification kit (Qiagen) was used to purify the transposed DNA, and libraries were amplified by PCR with barcoded Nextera primers (Illumina) using 2× NEB Next High-Fidelity PCR Master Mix. The libraries were purified, and sizes selected with AMPure XP beads (Beckman Coulter) for fragments between ~100 and 1000 bp in length according to the manufacturer's instructions. Paired-end sequencing was performed on an Illumina NextSeq 500 high-output flow cell.

### ATAC-seq data processing and data analysis
Sequence reads were mapped to the hg38 human reference genome using Bowtie 2.1.0[52], as previously described[53] Duplicate fragments and reads with mapping quality <10 was discarded. Differential peak calling was performed using DESeq2 (ver. 1.38.3)[39] using default parameters and requiring fold changes >1.5-fold in either direction or FDR-adjusted *P*-values < 0.10. To construct the PCA plot for the ATAC-seq data, we employed the DESeq2 package (ver. 1.38.3). For estimating the overlap between BED files, we utilized bedtools (ver. 2.28). Peak calling was performed by MACS 2.1.1[54]. Motif enrichment was measured using HOMER (version 4.11)[55].

### Chromatin immunoprecipitation (ChIP) followed by qPCR
Chromatin immunoprecipitation (ChIP) assays were performed at the Epigenomics Profiling Core at MD Anderson Cancer Center following the high throughput ChIP protocol with some modifications[56]. Briefly, cells were crosslinked in 1% formaldehyde, followed by incubation with glycine to stop crosslinking. Cells were collected and washed with ice-cold PBS and incubated with cell lysis buffer (5 mM PIPES pH 8.0, 85 mM KCl, and 0.5% NP-40) to isolate nuclei. The nuclei were lysed for 30 min on ice using nuclei lysis buffer (12 mM Tris−HCl pH 7.5, 6 mM EDTA pH 8.0, 0.5% SDS) supplemented with protease inhibitors. Lysates were fragmented with a Bioruptor (Diagenode) to obtain DNA fragments ranging from 200–600 bp. The supernatant was incubated with respective antibodies conjugated with Dynabeads Protein G (Invitrogen) overnight at 4 °C. The antibodies for SMARCB1 (#91735) was from Cell Signaling Tech and H3K27ac, H3K4me3 and H3 were from Abcam shown in Supplementary Table 3. The immunocomplexes were collected using Dynamag, washed extensively as described in the protocol, and reverse crosslinked overnight followed by DNA extraction. The DNA region of interest was detected by SYBR green real-time quantitative PCR (qPCR) using primers encompassing H3K27ac and SMARCB1 enrichment locus on human STAT3 promoter determined using previously published ChIP-Seq datasets[57]. The sequences of the forward and reverse primers used for ChIP-qPCR are described in Supplementary Table 5. Data was analyzed using GraphPad Prism v10.

### shRNA-mediated knockdown of SMARCB1 and STAT3
Lentiviruses for SMARCB1 knockdown and scrambled shRNA were obtained from Sigma, MO. Cells transfected were selected using puromycin selection. Lentiviruses for STAT3 knockdown and scrambled shRNA were synthesized by C-BASS core (BCM, USA), and lentiviral transduction was performed by manufacturer's instructions (Sigma, MO). The sequences for shRNAs are provided in Supplementary Table 6.

### TTI-101 treatment of orthotopic cell line-derived xenografts
SMARCB1 KO cells and Control cells were transplanted independently into the bladder wall of immune-deficient 6–8-week-old female NSG mice (Jackson Labs, USA). Once tumors were established, mice were randomized based on the BLI signal. Mice were treated with vehicle [60% Labrasol (Gattefosse, USA) with 40% PEG-400 (Selleckchem, USA)] or with TTI-101 (Obtained from Tvardi therapeutics) (50 mg/kg; Dissolved in 60% Labrasol with 40% PEG-400).

### Characterization of patient-derived xenografts (PDXs) and therapeutic experiments using TTI-101
TM00020 PDX obtained from Jackson Labs has SMARCB1 deletion (Supplementary Data 9). BLCA PDXs were originally derived by transplanting a fresh patient bladder tumor biopsy into the mammary gland fat pad of immunocompromised C.B-17/lcrHsd-Prkdc^scid^Lyst^bg-J^ (SCID BEIGE) mice. BLCA tumor fragments from TM00020 were implanted into male recipient mice with intact mammary fat pads while tumor fragments from BCM-BL8091 (Obtained from Baylor College of Medicine, PDX core) were implanted in female recipient mice with cleared mammary fat pads. Tumor samples (2 × 2 mm) were serially passaged

in SCID BEIGE mice by fat pad transplant under general anesthesia[58]. The weight of the mice was recorded, and tumor volumes were measured and calculated [$0.5 \times$ (long dimension) $\times$ (short dimension)$^2$] accordingly. When tumors reached an average size of 100–200 mm³, mice were randomized (in $\geq 5$ per group) and treatment with TTI-101 or vehicle control was initiated.

All mouse experiments were conducted with 3–5 mice per cage and maintained in a specific pathogen free environment with a-14-h light cycle/-10-h dark cycle, -20–23 °C temperature, -30–70% humidity, food (PicoLab® Select Rodent 50 IF/6 F) and water provided ad libitum. For all the mice experiments, tumor volumes 1500 mm³ or endpoint, the maximum allowed weight loss was 10% of total body weight and the maximum limit for tumor burden and weight loss was not exceeded in this study.

### Statistical analysis for in vitro and in vivo experiments

Two-sided statistical analysis or three-way comparison was considered statistically significant at $p < 0.05$. The statistical analysis was performed either in GraphPad Prism v10 for the graphs or with the R software for the heat maps. Error bars represent mean ± the standard error of the mean (SEM) or standard deviation (SD) wherever indicated.

### Identification of transcriptional signature from mouse xenografts and correlation with survival analysis in patients with BLCA

Among 393 differentially expressed genes with a fold change >2 and FDR < 0.05, we identified 59 upregulated and 334 downregulated. We have used 59 upregulated genes to further evaluate and identified that 55 gene data was available in TCGA-BLCA[7]. 55 out of 59 upregulated genes to predict patients with SMARCB1-deficiency (activation of pSTAT3) using TCGA-BLCA. For the classification model, we utilized the mRNA expression (RSEM-RNA-seq expectation maximalization) of 55 genes from TCGA-BLCA. Our classification model in patients with 1-sensitivity [True positive rate (TPR)] vs 1-specificity [False positive rate (FPR)] by Receiver Operating Characteristic (ROC) analysis using the logistic regression classifier. All statistical analysis was performed using R Statistical Software (version 4.1.1; R Foundation for Statistical Computing, Vienna, Austria).

### Reporting summary

Further information on research design is available in the Nature Portfolio Reporting Summary linked to this article.

## Data availability

All the raw data was available in Source Data. Raw and processed RNA-Seq data from mouse xenografts are available at GEO Hub with accession number GSE212762. High-throughput data generated in this study (ATAC-seq) have been deposited in the Gene Expression Omnibus (GEO) database with accession number GSE213964. Raw data and uncropped blots corresponding to Figs and Supplementary Figs are included in a zipped folder within the Source Data file or Supplementary data. Source data are provided with this paper.

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

## Acknowledgements

This research was fully supported by, NIH/NCI R01CA282282(NP) NIH/NCI R01CA220297 (NP) and NIH/NCI R01CA216426 (NP), PRCRP W81XWH-21-1-0613 (DOD award number CA200996P1) (NP), P20CA284971 (NP) and partially funded by NIH/other grant numbers, P42ES027725 (NP, SH), P30CA125123 Metabolomics shared resources (NP), CPRIT proteomics and metabolomics core facility (RP210227). This work received support from UTSW Simmons Comprehensive Cancer Center's Tissue Management Shared Resource and was supported by the National Cancer Institute of the National Institutes of Health under award number P30CA142543. This study was also partially supported by Susan G. Komen (CCR16380599 to SMK) and the DOD (W81XWH-18-1-0040 and W81XWH- 18-1-0084 to SMK) and CPRIT Core Facility Award (RP170691) to the Patient-derived Xenograft and Advanced In Vivo Models Core. This study was also partially supported by NIH S10 instrument award NIH S10OD028648 (SH) and R01CA272769 (HCH). This work is also partially supported by NIH/NCATS 1UL1TR003167 and the Cancer Prevention and Research Institute of Texas through grant RP170668 (WJZ). The study was also supported in part by the Cancer Center Support Grant to The University of Texas MD Anderson Cancer Center (grant number P30 CA016672) from the National Cancer Institute of the National Institutes of Health. PM was supported by a Career Development Award by the American Society of Clinical Oncology, a Research Award by KCCure, the MD Anderson Khalifa Scholar Award, the Andrew Sabin Family Foundation Fellowship, Gateway for Cancer Research, a Translational Research Partnership Award (KC200096P1) by the United States Department of Defense, an Advanced Discovery Award by the Kidney Cancer Association, a Translational Research Award by the V Foundation, the MD Anderson Physician-Scientist Award, and philanthropic donations by the Chris "CJ" Johnson Foundation and by the family of Mike and Mary Allen. We would like to thank Tvardi Therapeutics, Inc. for providing TTI-101 and guidance for therapeutic experiments. We would like to thank Dr. Abu Hena Mostafa Kamal, Ms. Vasanta Putluri, Mr. Chandra Shekar R. Ambati, Ms. He Rong for their technical support in performing in vitro and in vivo experiments. We would like to NCI Cooperative Human Tissue Network (CHTN) and thank Dr. Livia Eberlin from BCM for providing her support in obtaining the samples. We thank Mr. Michael Nguyen and Dr. Yuan Yao from Antibody-based proteomics core/shared resources for their excellent technical assistant in performing the RPPA experiments. We thank Dr. Cristian Coarfa from BCM for his support in RPPA data processing and

normalization. We would like to thank Dr. Abhinav Jain and the MDACC Epigenomics Profiling Core Facility for their assistance in the ChIP assays. We thank Dr. Fabio Stossi at BCM for his support and suggestions in microscope imaging. We would like to thank Advanced Cell Engineering and 3D Models (ACE-3M) Core, Human Tissue Acquisition and Pathology Core, and Integrated Microscopy Core at Baylor College of Medicine for their technical support.

## Author contributions

C.S.A.: Project conception, performed in vivo, in vitro experiments, analyzed data, full responsibility for the finished work and/or the conduct of the study, had access to the entire dataset, and manuscript writing. K.R.K.R.: Helped on in vivo and western blot experiments and provided his input on technical issues. Y.Y.: Developed algorithms, performed survival analysis for SMARCB1, RNA-seq, GSEA analysis, and analyzed data. Y.S.C.: Performed ATAC-seq library prep, analyzed ATAC-seq data, performed the GSEA analysis from TCGA and analyzed data. D.W.B.P.: Performed the ROC analysis for prediction of SMARCB1-deficient BLCA. L.E.D.: Assisted in the implantation of PDX for the project. D.J.H.S.: Developed algorithms, performed the survival analysis for SMARCB1. Z.S. and S.H.: Performed the RPPA analysis and provided the data. J.X.: Provided inputs on generation of knockdown cells and DNA sanger sequencing. M.J.E.: Edited the manuscript. A.B.A.: Provided RNA-seq data (Primary and matched metastatic) and edited the manuscript. L.Y.B.: Pathology scoring and identification of tumor cells in metastatic lesion. J.G.: Provided intellectual input and edited the manuscript. D.E.H.: Provided clinical input and edited the manuscript. Y.L.: Provided intellectual input and edited manuscript. H.C.H.: Provided input on ATAC-seq, intellectual input and supervision to Y.S.C. S.P.L.: Provided intellectual input, clinical input, helped to obtain the PDX from BCM PDX core and edited manuscript. C.J.C.: Provided the pan-cancer analysis, GSEA analysis, and edited the manuscript. A.S.: Provided intellectual input, mechanistic input, and edited manuscript. W.J.Z.: intellectual input, supervision to Y.Y. on survival analysis and RNA-seq, GSEA analysis. P.M.: helped on generation of KO cells, project conception, provided intellectual input, mechanistic input, edited the manuscript and controlled the decision to publish. S.M.K.: Project conception, intellectual input, manuscript writing, had access to the entire dataset and controlled the decision to publish. N.P.: Project conception, intellectual input, had access to the entire dataset, supervision, manuscript writing and controlled the decision to publish.

## Competing interests

S.M.K. is stakeholder of NeoZenome Therapeutics Inc. P.M. has received honoraria for service on a Scientific Advisory Board for Mirati Therapeutics, Bristol Myers Squibb, and Exelixis; consulting for Axiom Healthcare Strategies; non-branded educational programs supported by Exelixis and Pfizer; and research funding for clinical trials from Takeda, Bristol Myers Squibb, Mirati Therapeutics, Gateway for Cancer Research, and The University of Texas MD Anderson Cancer Center. M.J.E. reports grants from CPRIT and McNair Medical Foundation during the conduct of the study; personal fees from AstraZeneca outside the submitted work; in addition, M.J.E. has a patent for PAM50 issued, licensed, and with royalties paid from Veracyte. M.J.E. is current employee of AstraZeneca. H.C.H. is a consultant for Avenge Bio, Inc. A.S.K. is Scientific Advisor to Karkinos Health Care Pvt Ltd, India and unpaid visiting faculty to Sri Sathya Sai Institute for Higher Learning, India. The other authors declare that they have no competing interests.

## Additional information

[1]Department of Molecular and Cellular Biology, Baylor College of Medicine, Houston, TX 77030, USA. [2]Dan L. Duncan Comprehensive Cancer Center, Baylor College of Medicine, Houston, TX 77030, USA. [3]Mcwilliams School of Biomedical Informatics, University of Texas Health Science Center at Houston, Houston, TX 77030, USA. [4]Center for Precision Environmental Health, Baylor College of Medicine, Houston, TX 77030, USA. [5]Advanced Technology Cores, Baylor College of Medicine, Houston, TX 77030, USA. [6]Lester and Sue Smith Breast Center, Baylor College of Medicine, Houston, TX 77030, USA. [7]Department of Education, Innovation and Technology, Baylor College of Medicine, Houston, TX 77030, USA. [8]Genitourinary Malignancies Branch, Center for Cancer Research, National Cancer Institute, National Institutes of Health, Bethesda, MD 20892, USA. [9]Division of Pathology and Laboratory Medicine, The University of Texas MD Anderson Cancer Center, Houston, TX 77030, USA. [10]Department of Genitourinary Medical Oncology, The University of Texas MD Anderson Cancer Center, Houston, TX 77030, USA. [11]Department of Urology, University of Texas Southwestern Medical Center, Dallas, TX 75390, USA. [12]Department of Bioengineering, Rice University, Houston, TX 77005, USA. [13]Scott Department of Urology, Baylor College of Medicine, Houston, TX 77030, USA. [14]Department of Medicine, Baylor College of Medicine, Houston, TX 77030, USA. [15]Department of Genitourinary Medical Oncology, Division of Cancer Medicine, The University of Texas MD Anderson Cancer Center, Houston, TX 77030, USA. [16]Department of Translational Molecular Pathology, The University of Texas MD Anderson Cancer Center, Houston, TX 77030, USA. [17]David H. Koch Center for Applied Research of Genitourinary Cancers, The University of Texas, MD Anderson Cancer Center, Houston, TX 77030, USA. [18]These authors contributed equally: Karthik Reddy Kami Reddy, Yang Yuntao. ✉e-mail: pmsaouel@mdanderson.org; meghashyam.kavuri@bcm.edu; putluri@bcm.edu

