## [Peer Review File · Nature Communications]

REVIEWER COMMENTS

Reviewer #1 (Remarks to the Author):

Very well conducted study in a clinically relevant topic by an expert team evaluating the putative role of IL6/JAK/STAT3 pathway and possible therapeutic targets in SMARCB1-deficient bladder cancer, which represents a clinical need. The methodology looks appropriate, and the results are well described, supporting the conclusions. The manuscript with the figures is well written and is easy to follow. Would suggest addressing the following questions before potential publication:

1. Consider adding a brief hypothesis at the end of the introduction section
2. Figure 4A: should the second lane have STAT3 KD – (not +)?
3. Avoid phrasing “cancer patients” and rather write “patients with cancer”
4. Is it worth evaluating any of the other identified pathways via GSEA -Fig 1C- (in addition to IL6/JAK/STAT3) in SMARCB1-deficient bladder cancer?
5. Can SMARCB1 be affected by germline mutations?
6. What are the proposed concrete next steps from this nice work?

Reviewer #2 (Remarks to the Author):

Amara et al. report an interesting study identifying IL6/JAK/STAT3 signaling axis as a therapeutic vulnerability in SMARCB1-deficient bladder cancer. The authors used RNA expression to define deficiency and showed that 59-gene signature panel generated from SMARCB1 KO tumors predicted SMARCB1 deficiency in patients with bladder cancer. Through a series of loss and gain of function, they demonstrated increased IL6/JAK1/STAT3 signaling axis in vivo. Finally, they showed the interest of STAT3 selective inhibitor, TTI-101, in reducing tumor growth and metastasis in SMARCB1 KO orthotopic xenografts and a SMARCB1 deficient patient derived PDX model. Overall the idea that SMARCB1 loss lead to activation of STAT3 is not novel, however, identifying therapeutic vulnerabilities in these tumors is important. However, I have major concerns regarding the study design:

1) We refer to SMARCB1 deficient tumors as tumors with loss of SMARCB1 expression assessed by IHC following homozygous deletions, mutations or translocations. Herein, the authors defined SMARCB1 deficient tumors are those with lower expression using RNAseq, including 331 tumors among 406 bladder tumors. This is completely different from SMARCB1 deficient tumors (i.e. rhabdoid tumors, RMC) where SMARCB1 expression is completely lost at the protein level. In the cohorts assessed, authors should perform IHC staining to assess the correlation between RNA and protein. Authors should comment on why they consider the majority of bladder cancer as SMARCB1-deficient cancers? If the authors apply the same criteria to other cancers of TCGA, what is the proportion of samples considered as SMARCB1-deficient tumors?

2) The authors used one bladder cell model (T24 BLCA cell line). However, I have a concern regarding the model as the authors established KO models, while in bladder patients, there is no complete loss of SMARCB1 expression by IHC. Authors need to clarify this issue as reduced expression of SMARCB1 at the RNA level is different from complete loss of its expression. Can the authors perform mechanistic data dissecting the effect of SMARCB1 reduced expression as compared to KO experiments? In the latter one, we can speculate that there would be an effect on SWI/SNF genomic locations and SWI/SNF composition. However, it is unclear to me what is the effect following reduced expression of SMARCB1.

3) The ATAC-seq experiments were done in one T24. Experiments need to be confirmed in a second bladder cell line to be valid. Authors should also compare their data to other publically available data in rhabdoid tumors allowing to assess the specificity of SMARCB1-deficiency in bladder versus other smarcb1-deficient tumors defined molecularly.

4) The authors showed (Figure 1F) reduced RNA expression of SMARCB1 in metastatic samples as compared to primary tumors. Can the authors perform IHC in these samples to quantify the loss by IHC?

Reviewer #3 (Remarks to the Author):

The role of the SWI/SNF component SMARCB1 in bladder cancer was investigated. Low expression of SMARCB1 correlated with poor prognosis of bladder cancer patients in the TCGA dataset. Analyzes with bladder cancer cells harboring a CRISPR/Cas9-mediated knockout of SMARCB1 and knockout cells with reintroduction of a SMARCB1 expression vector showed that SMARCB1 interferes with invasiveness, migration and metastasis. RNA-seq studies showed that deletion of SMARCB1 activates Jak1/Stat3 signaling through improved accessibility of the Stat3 promoter. A bladder cancer subtype with reduced SMARCB1 and increased Stat3 activity has been identified that may be responsive to Stat3 inhibition.

General Comment

The data are interesting and experiments are state of the art but most of the data have already been published by Ding et al. (DOI: 10.1186/s12935-021-02363-3). These include the poor prognosis of bladder cancer patients with low SMARCB1 expression, the negative role of SMARCB1 in bladder cancer cell migration and proliferation, and increased Stat3 activation after SMARCB1 deletion. Ding et al. also showed that bladder cancer with low SMARCB1 expression might be sensitive to EGFR inhibition, but they did not perform therapeutic studies with Stat3 inhibitors.

Other Comments

1. The first heading and the first paragraph of the results should be deleted.
2. Are Figure 1D and E mixed up? Figure 1E looks more like GSEA of the TCGA data.
3. Why do the SMARCB1 knockout cells express SMARCB1 in Figure 3E?
4. Other Jak inhibitors should be used in Figure 3H to assess the role of Jak2 and Tyk2. Activating Jak1 phosphorylation should also be investigated.

Reviewer #4 (Remarks to the Author):

In the study by Amara et al., the authors investigate the effect of SMARCB1 deficiency in bladder cancer using published data, several genomics methods, and mouse models. Bladder cancer is characterized by mutation in genes involved in chromatin regulation, consisting of components of SWI/SNF complex and MLL complexes. However, the mutation and alteration in SMARCB1 is not very common and so far not very well studied. There is one study (Ding et al. 2021), which is also cited by the authors, which show the decreased patient survival with low SMARCB1 expression and implicating STAT3 as the main player in this regulatory scheme. In this regard, the main difference separating this study from Ding et al. is the extensive use of mouse models and the utility of STAT3 inhibitor, TTI-101 in tumor models. Although the study produced quite some data regarding preclinical aspects, there are several unclear points and missing information regarding the molecular and genomic analysis.

Major points:

- 1) In Figure 1B, it seems that majority of patients showing deletion belongs to the group 'shallow deletion'. Could the authors test whether the comparison between 'diploid' and 'shallow deletion' group is significant? It might be good to add an oncoprint image showing the overall mutation and genomic alteration frequency of SMARCB1 in bladder cancer.
- 2) For some of the figures, it is not clear, whether the used replicates are biological or technical. Could the authors provide this information for all the figures? For instance, are the replicates for ATAC-seq biological or technical? Additionally, name of the statistical test used for each analysis should be written in the legends of all related figure panels. Example: In Figure 1B, the used test information is missing.
- 3) Sanger sequencing result presented for SMARCB1 knockout in Supplementary Figure 2 is not clean. Could the authors provide a cleaner sequencing result?
- 4) The authors performed ATAC-seq in SMARCB1 KO, control and rescue cell lines. The results of ATAC-seq is only shown for the loci IL6 and STAT3. However, there is no data provided for the genome-wide analysis of ATAC-seq data except the sample clustering and PCA plot. For instance, how many peaks were identified, what are the overlaps between different conditions, what is the

relationship with the identified gene expression changes ? In the methods section for ATAC-seq part, it is written that transcription factor motif enrichment was done with HOMER, but there is no analysis presented for transcription factor motif finding, either.

5) The authors performed H3K27ac and H3K4me3 ChIP-qPCR at STAT3 to show the chromatin accessibility at this gene and relate to its expression status. The authors comment that SMARCB1 may repress STAT3 expression. As SMARCB1 is normally expected to be involved in active chromatin organization, to prove this claim authors could perform ChIP-qPCR for SMARCB1 at STAT3 gene.

6) In Figure 4A, in lane 2 and lane 4, exactly the same conditions are used (SMARCB1 KO and STAT3 kd). However, although there is a substantial decrease in STAT3 level in lane 4, this is not the case for lane 2. In figure legend, it is written that lane 3 and lane 4 were used for the in vivo experiments. Could the authors explain the issue with lane 2 ? Is it a typo in labeling of the experiment ?

7) In the last part of the results section, the authors identify a transcriptional signature related to SMARCB1 loss. It is written that this was based on the expression changes of genes dysregulated in SMARCB1 loss and and restoration in T24 cells. As the authors did the expression analysis in xenografts but not in cell lines directly, this point might be clarified in the text. Overall, this analysis and results might better fit to the results presented in Figure 3, where the authors first describe the changes in transcriptome in xenograft models. Additionally, there seems some problems with the results presented in Supplementary Figure 12, for the identification of this transcriptomic signature. The authors use two different methods for the analysis of differentially expressed genes (Salmon and XenofilteR). In Supplementary Figure 12A and 12B, number of genes identified using two different methods are exactly the same, further the heatmaps look very similar, too. Similarly, Supplementary Figure 12C has the same problem, too. Could the authors comment on this issue ?

Minor points:

- Manuscript could be double-checked for the appearance of parentheses, double dots, etc.
- Reference to Figure 6B is missing in the text.
- Legend of Supplementary Figure 7 could be checked. A-B panels referred as GSEA plots, but they are actually volcano plots. The text for E-F panels could be clarified.

RESPONSE TO REVIEWER COMMENTS.

Reviewer #1 (Remarks to the Author):

Very well conducted study in a clinically relevant topic by an expert team evaluating the putative role of IL6/JAK/STAT3 pathway and possible therapeutic targets in SMARCB1-deficient bladder cancer, which represents a clinical need. The methodology looks appropriate, and the results are well described, supporting the conclusions. The manuscript with the figures is well written and is easy to follow. Would suggest addressing the following questions before potential publication:

1. Consider adding a brief hypothesis at the end of the introduction section.

Response: We appreciate the reviewer's feedback. In the revised manuscript, we have now accordingly included the brief hypothesis at the end of the introduction section.

2. Figure 4A: should the second lane have STAT3 KD – (not +)?

Response: We have corrected this typo in **Fig. 4A**.

3. Avoid phrasing "cancer patients" and rather write "patients with cancer"

Response: We have accordingly changed "cancer patients" to "patients with cancer" throughout the manuscript.

4. Is it worth evaluating any of the other identified pathways via GSEA -Fig 1C- (in addition to IL6/JAK/STAT3) in SMARCB1-deficient bladder cancer?

Response: We appreciate the reviewer's suggestion. While beyond the scope of the current manuscript, we have now noted in the discussion section (Lines 346-355) that further research is warranted in pathways beyond the IL-6 / JAK / STAT3 that are deregulated in SMARCB1-deficient bladder cancer. As a reference to guide future efforts, the additionally identified pathways via GSEA are now listed in **Supplementary Table 2 and Supplementary Table 6**.

5. Can SMARCB1 be affected by germline mutations?

Response:

We appreciate the insightful comment. An analysis of pathogenic germline variants in over 10,389 cancers (PMID: 29625052) identified one patient with ovarian cancer and a SMARCB1 mutation. In addition, up to 20% of malignant rhabdoid tumors have germline mutations (PMID: 29280680). However, no association between SMARCB1 germline mutations and bladder cancer has been reported to date. We have included this observation in the manuscript (Lines 330-332). Future studies should investigate the possibility that polygenic germline interactions may influence SMARCB1 expression in urothelial carcinomas. We have revised the discussion section to include these considerations.

6. What are the proposed concrete next steps from this nice work?

Response: Our analyses of human samples suggest that unbiased CLIA and CAP-accredited tumor genomic and transcriptomic assays can be used to also detect SMARCB1-deficient bladder cancer, defined as tumors with low SMARCB1 mRNA expression and concomitant deep or shallow SMARCB1 deletions. Our reported gene signature of SMARCB1 deficiency in bladder cancer can help guide these efforts. Our study suggests that the clinical utility of detecting this subset is that it may prompt subsequent testing for upregulation and potential targeting of the STAT3 signaling axis. This has now been addressed in the revised discussion section.

Reviewer #2 (Remarks to the Author):

Amara et al. report an interesting study identifying IL6/JAK/STAT3 signaling axis as a therapeutic vulnerability in SMARCB1-deficient bladder cancer. The authors used RNA expression to define deficiency and showed that 59-gene signature panel generated from SMARCB1 KO tumors predicted SMARCB1 deficiency in patients with bladder cancer. Through a series of loss and gain of function, they demonstrated increased IL6/JAK1/STAT3 signaling axis in vivo. Finally, they showed the interest of STAT3 selective inhibitor, TTI-101, in reducing tumor growth and metastasis in SMARCB1 KO orthotopic xenografts and a SMARCB1-deficient patient derived PDX model. Overall the idea that SMARCB1 loss lead to activation of STAT3 is not novel, however, identifying therapeutic vulnerabilities in these tumors is important. However, I have major concerns regarding the study design:

1) We refer to SMARCB1 deficient tumors as tumors with loss of SMARCB1 expression assessed by IHC following homozygous deletions, mutations or translocations. Herein, the authors defined SMARCB1 deficient tumors are those with lower expression using RNAseq, including 331 tumors among 406 bladder tumors. This is completely different from SMARCB1 deficient tumors (i.e. rhabdoid tumors, RMC) where SMARCB1 expression is completely lost at the protein level. In the cohorts assessed, authors should perform IHC staining to assess the correlation between RNA and protein. Authors should comment on why they consider the majority of bladder cancer as SMARCB1-deficient cancers? If the authors apply the same criteria to other cancers of TCGA, what is the proportion of samples considered as SMARCB1-deficient tumors?

Response: We appreciate the reviewer's careful review of the manuscript and insightful comments. We defined in our manuscript "SMARCB1-deficient" BLCA tumors as those having low SMARCB1 RNA expression and shallow or deep SMARCB1 deletions. By this definition, ~32% of BLCA tumors are considered to be SMARCB1-deficient (**Fig. 1E** of the manuscript). Note that our study necessitated a refinement of the conventional terminology that assumes that SMARCB1-deficient tumors are only those with complete loss of SMARCB1 expression such as rhabdoid tumors and RMC. Instead, we now show in our revised results that bladder cancer cells expressing reduced levels of SMARCB1 can also upregulate the IL6/JAK/STAT3 signaling axis similarly to SMARCB1 negative bladder cancer cells conferring functional equivalence and a therapeutic sensitivity to STAT3 inhibition. Thus, our results suggest that both SMARCB1-null and SMARCB1-low bladder cancer can be SMARCB1-deficient, at least with regards to specific pathways such as the IL6/JAK/STAT3 signaling axis. This further opens new avenues to explore other commonalities between SMARCB1-null and SMARCB1-low bladder cancers, beyond the scope of

our manuscript. This has now been further clarified in the discussion section. In addition, per the reviewer's recommendation, we have provided the proportions of SMARCB1-deficient tumors in non-BLCA samples from TCGA (Refer to **Supplementary Figure 2F**; **Supplementary Table 5**). We assessed the correlation between SMARCB1 mRNA by qPCR and protein levels by western blot analysis (Quantified by Image J) in an CHTN (Cooperative human tissue network) BLCA cohort obtained by MTA (Material transfer agreement upon IRB approval H-35808) (**Figure RR1**). Furthermore, in the established TM00020 BLCA PDX (**Supplementary Table 8**) model known to harbor SMARCB1 deletion and low mRNA, we performed IHC showing decreased SMARCB1 protein expression compared to the SMARCB1 proficient BL8091 BLCA PDX model (**Supplementary Figure 13A-C**).

Figure RR1: Correlation of SMARCB1 protein and mRNA expression levels from independent cohort of BLCA specimens (n=27). Protein expression was quantified (background signal was used to subtract) by Image J and normalized with β -actin signal. Correlation was plotted using GraphPad Prism v10. (Note: SMARCB1 and β -actin primers were listed in Supplementary Table 10).

2) The authors used one bladder cell model (T24 BLCA cell line). However, I have a concern regarding the model as the authors established KO models, while in bladder patients, there is no complete loss of SMARCB1 expression by IHC. Authors need to clarify this issue as reduced expression of SMARCB1 at the RNA level is different from complete loss of its expression. Can the authors perform mechanistic data dissecting the effect of SMARCB1 reduced expression as compared to KO experiments? In the latter one, we can speculate that there would be an effect on SWI/SNF genomic locations and SWI/SNF composition. However, it is unclear to me what is the effect following reduced expression of SMARCB1.

Response: Per the reviewer's suggestion, we have now used two independent shRNAs targeting SMARCB1 to generate two separate T24 SMARCB1 knockdown cell models, confirmed that both demonstrated reduced (but not negative) expression of SMARCB1 and compared the effect of these shSMARCB1 versus SMARCB1 KO cells on tumor growth. In the revised manuscript, we show that reduced expression of SMARCB1 (shSMARCB1) again resulted significantly increased tumor growth as compared to the control tumors, suggesting that reduced expression of SMARCB1 (shSMARCB1) is sufficient to induce tumor growth in these BLCA models (**Revised Supplementary Figure 4E-G**). We additionally observed that SMARCB1-KO cells formed tumors within 29 days as compared to Sh-SMARCB1 which formed tumors in 59 days, suggesting that complete loss of SMARCB1 BLCA results in more aggressive in a dose-dependent effect consistent with the notion that "SMARCB1-deficiency" in BLCA is a gradient as opposed to the binary notion "loss" versus "presence". In addition, we isolated the proteins from these tumors and measured the levels of pSTAT3, STAT3 levels by western blot and IL6 levels by ELISA. Our results revealed strong negative correlations of SMARCB1 levels with IL6/STAT3 signaling (**Revised Supplementary Figure 9F-G**). Lastly, we performed targeted CHIP-qPCR in the SMARCB1 knockdown setting (SMARCB1 KD2 T24

cell line) and observed increased accessibility of H3K27Ac and H3K4Me3 peaks in SMARCB1 knockdown tumors compared to shCtrl (**Supplementary Figure 10G**).

3) The ATAC-seq experiments were done in one T24. Experiments need to be confirmed in a second bladder cell line to be valid. Authors should also compare their data to other publically available data in rhabdoid tumors allowing to assess the specificity of SMARCB1-deficiency in bladder versus other smarcb1-deficient tumors defined molecularly

Response: As suggested by the reviewer, we performed SMARCB1 knockdown experiments using two separate shRNA constructs in the UPPL1541 murine bladder cancer cell line. We then performed ATAC-seq and hypergeometric test on the nearest genes to the peaks with increased accessibility upon SMARCB1 knockdown by shRNA (KD2) and found that the

Figure RR2: ATAC-seq in UPPL1541 SMARCB1 knockdown shows increased accessibility of HALLMARK_IL6_JAK_STAT3 signaling pathway. **A)** Western blot analysis of SMARCB1 and β -actin in UPPL1541 SMARCB1 knockdown cells. shCtrl (n=2) and SMARCB1 kd2 (n=2) were used for performing ATAC-seq and pathway enrichment analysis. **B)** Ranking of hallmark gene sets that have increased accessibility in ATAC-seq upon knockdown of SMARCB1 in UPPL1541 cell line. HALLMARK_IL6_JAK_STAT3 signaling pathway was indicated by Red arrow. **C)** Genome track demonstrating increased accessibility of the overlapped genes of HALLMARK IL6_JAK_STAT3 signaling pathway (TNF, IL1R1, CSF3R, CNTFR and JUN) between T24

Hallmark_Il6_JAK_STAT3_signaling pathway was significantly enriched by SMARCB1 knockdown (**Figure RR2**).

In addition, we compared the ATAC-seq data shown in [Revised Supplementary Figure 10E-F] for STAT3 and IL6 loci from ATAC-seq of the rhabdoid tumor cell line TTC1240 with or without SMARCB1 re-expression. Our results revealed that SMARCB1 re-expression in rhabdoid tumors (GSE124903) showed no changes in the accessibility of IL6 and STAT3 loci in this cell line model (Figure RR3).

4) The authors showed (Figure 1F) reduced RNA expression of SMARCB1 in metastatic samples as compared to primary tumors. Can the authors perform IHC in these samples to quantify the loss by IHC?

Response: We would like to thank the reviewer for this comment. Unfortunately, we only have access to the RNA sequencing data as these patient tumor samples have been depleted. However, as we described above, there is a moderate correlation between SMARCB1 RNA expression and protein levels in both human bladder cancer samples (Figure RR1) and PDX models (Supplementary Figure 13).

Reviewer #3 (Remarks to the Author):

The data are interesting and experiments are state of the art but most of the data have already been published by Ding et al. (DOI: 10.1186/s12935-021-02363-3). These include the poor prognosis of bladder cancer patients with low SMARCB1 expression, the negative role of SMARCB1 in bladder cancer cell migration and proliferation, and increased Stat3 activation after SMARCB1 deletion. Ding et al. also showed that bladder cancer with low SMARCB1 expression might be sensitive to EGFR inhibition, but they did not perform therapeutic studies with Stat3 inhibitors.

Response: We would like to thank the reviewer for allowing us to highlight the novelty of our mechanistic and therapeutic studies. We comprehensively define and show in our revised manuscript the impact of SMARCB1 deficiency on tumor growth, STAT3 phosphorylation, STAT3 inhibitor susceptibility, and STAT3 occupancy. Our study has provided new *in vitro* and *in vivo* experimental and mechanistic data dissecting the effect of SMARCB1 deficiency in BLCA which was not investigated by Ding et al. In addition, we have provided multiple levels of orthogonal *in vitro* and *in vivo* evidence showing the activation of pSTAT3 through IL6-JAK signaling. Furthermore, as the reviewer noted, we performed for the first time *in vitro* and *in vivo* therapeutic experiments of pSTAT3 inhibition in both a SMARCB1 knock out orthotopic BLCA xenograft model (**Fig. 5**) and a bladder cancer PDX model naturally harboring low SMARCB1 levels but not complete SMARCB1 loss (**Fig. 6**). This allowed us to comprehensively evaluate the efficacy of pSTAT3 inhibition across different levels of SMARCB1 deficiency (low SMARCB1 expression associated with *SMARCB1* deletion).

Other Comments

1. The first heading and the first paragraph of the results should be deleted.

Response: We have accordingly now deleted the first paragraph of the results section (Refer to Lines 112-114 in the revised manuscript).

2. Are Figure 1D and E mixed up? Figure 1E looks more like GSEA of the TCGA data.

Response: We have revised Figure 1 and added further GSEA analyses using independent cohorts (GSE31684; GSE32548; GSE48276) and included in the **Supplementary Figure 2A-C**.

3. Why do the SMARCB1 knockout cells express SMARCB1 in Figure 3E?

Response: Due to the high homology of SMARCB1 between human and mouse (nearly ~99%), we detected SMARCB1 levels in the orthotopic SMARCB1-knockout tumors isolated from mouse bladders because the antibody also picks the SMARCB1 protein from mouse tissues (**Refer to Supplementary Figure 6B**).

4. Other Jak inhibitors should be used in Figure 3H to assess the role of Jak2 and Tyk2. Activating Jak1 phosphorylation should also be investigated.

Response: As suggested by the reviewer, we have now treated spheroids from T24 Control, SMARCB1 knockout and SMARCB1 rescue cells with JAK2 inhibitor AZ960 (PMID: 18775810) and TYK2 inhibitor Deucravacitinib (PMID: 31318208). We found that inhibiting JAK2 and TYK2 lead to downregulation of pSTAT3 Y705 phosphorylation as shown in **Figure RR4**. Furthermore, due to low signal intensities observed with immunoblot for pJAK1, we performed RPPA in T24 spheroids which confirmed a significant

increase in the phosphorylated active form of JAK1 (pJAK1) in SMARCB1 KO compared with control and SMARCB1 rescue (**Supplementary Figure 9D**).

Reviewer #4 (Remarks to the Author):

1) In Figure 1B, it seems that majority of patients showing deletion belongs to the group 'shallow deletion'. Could the authors test whether the comparison between 'diploid' and 'shallow deletion' group is significant? It might be good to add an oncoprint image showing the overall mutation and genomic alteration frequency of SMARCB1 in bladder cancer.

Response: As suggested by the reviewer, we performed expression analysis comparing SMARCB1 diploid vs shallow deletion and show that the SMARCB1 expression is significantly lower in SMARCB1 deletion patients as compared to diploid bladder cancer. Furthermore, we found that pSTAT3 (Y705) was significantly higher in the SMARCB1 deletion subgroups (**Revised Figure 1D-E**).

As requested by the reviewer, we have now provided the oncoprint of SMARCB1 in TCGA BLCA tumors (**Supplementary Figure S2E**).

2) For some of the figures, it is not clear, whether the used replicates are biological or technical. Could the authors provide this information for all the figures? For instance, are the replicates for ATAC-seq biological or technical? Additionally, name of the statistical test used for each analysis should be written in the legends of all related figure panels. Example: In Figure 1B, the used test information is missing.

Response: We appreciate the reviewer's careful review of the manuscript. The replicates for ATAC-seq are two biological replicates and these are indicated in lines 965-967. We also now provide the type of replicate information and statistical test used for all related figures.

3) Sanger sequencing result presented for SMARCB1 knockout in Supplementary Figure 2 is not clean. Could the authors provide a cleaner sequencing result?

Response: We have repeated the Sanger sequencing and have now provided a cleaner sequencing result in **Supplementary Figure 3**.

4) The authors performed ATAC-seq in SMARCB1 KO, control and rescue cell lines. The results of ATAC-seq is only shown for the loci IL6 and STAT3. However, there is no data provided for the genome-wide analysis of ATAC-seq data except the sample clustering and PCA plot. For instance, how many peaks were identified, what are the overlaps between different conditions, what is the relationship with the identified gene expression changes? In the methods section for ATAC-seq part, it is written that transcription factor motif enrichment was done with HOMER, but there is no analysis presented for transcription factor motif finding, either.

Response: As suggested by the reviewer, we have analyzed the number of peaks in each condition and analyzed the overlaps between different conditions. These results are now presented in **Supplementary Figure 10A-F**. In summary, SMARCB1 KO in T24 cells leads to loss of accessibility in 60124 peaks and 53952

of those peaks has regained accessibility when SMARCB1 is re-expressed. SMARCB1 KO in T24 cells leads to gained accessibility in 38313 peaks and 30712 of those peaks lose the accessibility when SMARCB1 is re-expressed, suggesting the accessibility of those peaks is regulated by SMARCB1.

We performed hypergeometric tests on the nearest genes to the peaks with increased accessibility upon SMARCB1 KO and showed that Hallmark_IL6_JAK_STAT3_signaling pathway was significantly enriched by SMARCB1 KO (**Supplementary Figure 10C**).

We now also include the HOMER motif analysis demonstrating significant enrichment of the STAT3 motif (**Supplementary Figure 10D**), demonstrating that the STAT3 motif is significantly enriched in the peaks with increased accessibility [($p < 0.001$)].

5) The authors performed H3K27ac and H3K4me3 ChIP-qPCR at STAT3 to show the chromatin accessibility at this gene and relate to its expression status. The authors comment that SMARCB1 may repress STAT3 expression. As SMARCB1 is normally expected to be involved in active chromatin organization, to prove this claim authors could perform ChIP-qPCR for SMARCB1 at STAT3 gene.

Response: As suggested by the reviewer, we performed ChIP-qPCR analysis for SMARCB1 in T24 Control, SMARCB1 KO and SMARCB1 rescue xenografts and observed decreased enrichment of SMARCB1 at the STAT3 locus in SMARCB1 knockout xenografts and increased enrichment of SMARCB1 at STAT3 locus in the SMARCB1 rescue settings (**Fig. 3F**).

6) In Figure 4A, in lane 2 and lane 4, exactly the same conditions are used (SMARCB1 KO and STAT3 kd). However, although there is a substantial decrease in STAT3 level in lane 4, this is not the case for lane 2. In figure legend, it is written that lane 3 and lane 4 were used for the in vivo experiments. Could the authors explain the issue with lane 2? Is it a typo in labeling of the experiment?

Response: We appreciate the reviewer's careful review and apologize for this labeling typo which has now been corrected in the revised version (**Fig. 4A**).

7) In the last part of the results section, the authors identify a transcriptional signature related to SMARCB1 loss. It is written that this was based on the expression changes of genes dysregulated in SMARCB1 loss and and restoration in T24 cells. As the authors did the expression analysis in xenografts but not in cell lines directly, this point might be clarified in the text. Overall, this analysis and results might better fit to the results presented in Figure 3, where the authors first describe the changes in transcriptome in xenograft models. Additionally, there seems some problems with the results presented in Supplementary Figure 12, for the identification of this transcriptomic signature. The authors use two different methods for the analysis of differentially expressed genes (Salmon and XenofilteR). In Supplementary Figure 12A and 12B, number of genes identified using two different methods are exactly the same, further the heatmaps look very similar, too. Similarly, Supplementary Figure 12C has the same problem, too. Could the authors comment on this issue?

Response: To mitigate the effect of mouse mRNA contamination and estimate mouse contamination rates, we first used the Salmon and the XenofilteR methods to obtain differentially expressed genes and

differential gene sets. Using the Salmon method, we have identified 1926 differentially expressed genes while comparing Control vs. KO, and 531 genes while comparing KO vs. Rescue. For XenofilteR method, we identified 2594 differentially expressed genes while comparing Control vs. KO, and 796 differentially expressed genes by comparing KO vs. Rescue. In the initial submission, we have represented common differentially expressed genes between the two methods, thus the heatmaps looks similar. In the revised version, we have included all differentially expressed genes per method basis including Salmon and XenofilteR. So, two methods identified different number of genes. These differentially expressed genes were represented in new **Supplementary Figure 14A-D**.

We only selected differentially expressed genes that were overlapping and gene sets identified by both methods for further analysis, as described in the RNA-seq and GSEA analysis Section in Methods. Therefore, 1537 differentially expressed genes for Control vs. KO and 444 genes for KO vs. Rescue are observed to be overlapping and were represented as in **Supplementary Figure 14E** and below Table 1 .

Table 1. Differentially expressed genes identified by different methods		
	Control vs. KO	KO vs. Rescue
Salmon	1926	531
XenofilteR	2594	796
Overlapped genes	1537	444

The 393 genes in **Supplementary Table 9** are the rewired genes (Identified from the overlapped genes) between two groups. As a representative heatmap we have presented one heatmap in **Supplementary Figure 14F** for 393 rewired genes by comparing SMARCB1 control and KO. As suggested by the reviewer to avoid repetition of the heatmap the rest of the comparisons are shown in **Supplementary Table 9**.

The rewired genes are defined as: (1) genes both up-regulated in T24 Control EV vs. SMARCB1 KO and down-regulated in SMARCB1 KO vs. SMARCB1 Rescue, or (2) genes both down-regulated in T24 Control EV vs. SMARCB1 KO and up-regulated genes in SMARCB1 KO vs. SMARCB1 Rescue. This has now been clarified in the revised version of the manuscript.

Rewired genes:

Minor points:

- Manuscript could be double-checked for the appearance of parentheses, double dots, etc.

Response: As suggested by the reviewers, we have carefully reviewed the manuscript and corrected typos such as misplaced parentheses and double dots.

- Reference to Figure 6B is missing in the text.

Response: We appreciate the reviewers for pointing out this error. We have now referenced **Fig. 6B** in the revised manuscript (Lines 297-298).

- Legend of Supplementary Figure 7 could be checked. A-B panels referred to as GSEA plots, but they are actually volcano plots. The text for E-F panels could be clarified.

Response: We have corrected this error in the revised manuscript (Lines 934-943; Refer to Supplementary Figure 8).

REVIEWERS' COMMENTS

Reviewer #1 (Remarks to the Author):

My comments were addressed, thank you

Reviewer #2 (Remarks to the Author):

The authors have to be congratulated on the amount of work done and on highly improving the current manuscript. No other comments. This manuscript would be a great addition to the field.

Reviewer #3 (Remarks to the Author):

The authors have improved the manuscript, but the lack of novelty remains, at least in my opinion. Most of the data have been published by Ding et al. (DOI: 10.1186/s12935-021-02363-3). Ding did not show chromatin data, but it is somehow obvious that a SWI/SNF component like SMARCB1 controls the IL6/STAT3 pathway via chromatin remodeling. The authors claim that “we performed for the first time in vitro and in vivo therapeutic experiments of pSTAT3 inhibition in both a SMARCB1 knock out orthotopic BLCA xenograft model (Fig. 5) and a bladder cancer PDX model naturally harboring low SMARCB1 levels but not complete SMARCB1 loss (Fig. 6)”. This is only partially true as Ding et al have already performed in vitro experiments with a STAT3 inhibitor demonstrating that STAT3 inhibition reduces colony formation and transwell migration of T24 and 5637 cells with SMARCB1 depletion. What is appreciated are the in vivo experiments indicating therapeutic efficacy of STAT3 inhibition in SMARCB1-low bladder cancer.

Reviewer #4 (Remarks to the Author):

Amara et al. submitted a significantly revised version of the manuscript, where they incorporated new experimental data and computational analysis. All of my comments are addressed and I think the study is suitable for publication in Nature Communications.

I have just a final comment regarding the transcription factor motif analysis presented in Supplementary Figure 10D. In the figure, only the enrichment of STAT3 and associated p-value is shown. However, to be complete, full list of transcription factors which are identified to be enriched can be presented or alternatively the authors might comment on the rank of STAT3 in HOMER transcription factor analysis.

RESPONSE TO REVIEWER COMMENTS

Reviewer #1 (Remarks to the Author):

My comments were addressed, thank you.

Response: We appreciate the reviewers' feedback and the opportunity to revise our manuscript.

Reviewer #2 (Remarks to the Author):

The authors have to be congratulated on the amount of work done and on highly improving the current manuscript. No other comments. This manuscript would be a great addition to the field.

Response: We thank the reviewer for their feedback and appreciate the opportunity to revise our manuscript.

Reviewer #3 (Remarks to the Author):

The authors have improved the manuscript, but the lack of novelty remains, at least in my opinion. Most of the data have been published by Ding et al. (DOI: 10.1186/s12935-021-02363-3). Ding did not show chromatin data, but it is somehow obvious that a SWI/SNF component like SMARCB1 controls the IL6/STAT3 pathway via chromatin remodeling. The authors claim that “we performed for the first time in vitro and in vivo therapeutic experiments of pSTAT3 inhibition in both a SMARCB1 knock out orthotopic BLCA xenograft model (Fig. 5) and a bladder cancer PDX model naturally harboring low SMARCB1 levels but not complete SMARCB1 loss (Fig. 6)”. This is only partially true as Ding et al have already performed in vitro experiments with a STAT3 inhibitor demonstrating that STAT3 inhibition reduces colony formation and transwell migration of T24 and 5637 cells with SMARCB1 depletion. What is appreciated are the in vivo experiments indicating therapeutic efficacy of STAT3 inhibition in SMARCB1-low bladder cancer.

Response: We appreciate the reviewer's feedback. It is indeed correct that our study provides for the first time comprehensive in vivo therapeutic and functional experiments, including pSTAT3 inhibition. We also note that Ding et al. (PMID: 34876150; already cited in our paper) did not use a pSTAT3 inhibitor *in vitro*, such as TTI-101 used in our study (IC₅₀ of 3.7 μM for inhibiting G-CSF-induced pSTAT3 levels as shown in PMID: 27027445) but rather an inhibitor of STAT3 binding (S3I-201; IC₅₀ of 86 μM for inhibition of STAT3 DNA binding as shown in PMID: 27027445). Regardless, we have carefully avoided such priority claims in the revised manuscript.

Reviewer #4 (Remarks to the Author):

Amara et al. submitted a significantly revised version of the manuscript, where they incorporated new experimental data and computational analysis. All of my comments are addressed and I think the study is suitable for publication in Nature Communications.

I have just a final comment regarding the transcription factor motif analysis presented in Supplementary Figure 10D. In the figure, only the enrichment of STAT3 and associated p-value is shown. However, to be complete, full list of transcription factors which are identified to be enriched can be presented or alternatively the authors might comment on the rank of STAT3 in HOMER transcription factor analysis.

Response: As suggested by the reviewer we have now provided the full list of increased motifs compared to unchanged sites upon SMARCB1 KO in T24 cell line as Supplementary Table 8.